# Newton Meets Marchenko-Pastur: Massively Parallel Second-Order Optimization with Hessian Sketching and Debiasing

**Elad Romanov, Fangzhao Zhang, and Mert Pilanci**
Stanford University
`{eromanov,zfzhao,pilanci}@stanford.edu`

## Abstract

Motivated by recent advances in serverless cloud computing, in particular the "function as a service" (FaaS) model, we consider the problem of minimizing a convex function in a massively parallel fashion, where communication between workers is limited. Focusing on the case of a twice-differentiable objective subject to an L2 penalty, we propose a scheme where the central node (server) effectively runs a Newton method, offloading its high per-iteration cost—stemming from the need to invert the Hessian—to the workers. In our solution, workers produce independently coarse but low-bias estimates of the inverse Hessian, using an adaptive sketching scheme. The server then averages the descent directions produced by the workers, yielding a good approximation for the exact Newton step. The main component of our adaptive sketching scheme is a low-complexity procedure for selecting the sketching dimension, an issue that was left largely unaddressed in the existing literature on Hessian sketching for distributed optimization. Our solution is based on ideas from asymptotic random matrix theory, specifically the Marchenko-Pastur law. For Gaussian sketching matrices, we derive non asymptotic guarantees for our algorithm which do not depend on the condition number of the Hessian nor a priori require the sketching dimension to be proportional to the dimension, as is often the case in asymptotic random matrix theory. Lastly, when the objective is self-concordant, we provide convergence guarantees for the approximate Newton's method with noisy Hessians, which may be of independent interest beyond the setting considered in this paper.

## 1 Introduction

Consider minimizing a convex, twice-differentiable function $F : \mathbb{R}^d \to \mathbb{R}$, subject to an L2 regularization penalty:

$$\min_{\theta \in \mathbb{R}^d} F(\theta) + \frac{\lambda}{2} \|\theta\|^2, \tag{1}$$

where $\lambda > 0$. Such problems frequently appear in machine learning and statistics. Common examples include ridge regression, logistic regression with regularization, support vector machines (SVMs) and kernel machines, among others (Hastie et al., 2009).

Consider a scenario where a large number of workers (processors) are available to collaboratively solve (1), with the assumptions that each worker (i) has full access to the objective function; (ii) operates with limited individual computational resources; and (iii) cannot communicate directly with other workers, only with a central server, with the latter responsible for orchestrating the work of the workers (a "star" network topology). This setting is motivated by recent developments in serverless cloud computing, particularly the "function as a service" (FaaS) model (Jonas et al., 2017). Serverless computing and FaaS are well-suited for computing gradients and Hessians on large datasets, enabling parallelization of these computations across a large number of workers. Additionally, this approach provides resilience to failed or straggler workers, ensuring robust and efficient processing (Bartan & Pilanci, 2019). In designing a solution of this setting, we are essentially guided by two key principles: (a) minimizing the number of communication or worker deployment rounds, with

each round utilizing a large number of workers in parallel; and (b) reducing the computational load on individual workers by distributing smaller tasks across many workers concurrently.

This paper proposes and analyzes a scheme for solving (1) in a massively parallel manner. In our scheme, the server aims to solve (1) using Newton's method, a second-order iterative algorithm which is known to converge extremely fast for sufficiently smooth and strongly convex objectives (quadratic convergence rate). Its fast convergence is attained by incorporating curvature (Hessian) information when searching for a descent direction, and specifically requires an inverse Hessian-gradient product at every iteration. This a priori results in massive per-iteration computational cost: exactly computing the descent direction (assuming the Hessian is accessible) costs practically $O(d^3)$ flops; in high-dimensional problems (large $d$), this may be prohibitively expensive.

We wish to offset the high per-iteration cost of Newton's method, by deploying many workers in parallel to collaboratively compute the Newton direction. In the setting we consider, workers cannot easily communicate with one another, and therefore directly inverting the Hessian by is not feasible. In our scheme, each worker independently computes a coarse but **unbiased** estimate of the exact Newton direction, using randomized sketching. These estimates are then aggregated (averaged) at the server, producing a good approximation of the true Newton step.

**Newton's method:**    The exact Newton method for minimizing (1) has the following form:

$$\theta_t = \theta_{t-1} - \alpha_t W_t g_t, \qquad (t = 1, \ldots, T), \tag{2}$$

where $\alpha_t$ is a step size (typically chosen by line search) and the gradient and inverse Hessian are

$$g_t := \nabla F(\theta_{t-1}) + \lambda \theta_{t-1}, \tag{3}$$

$$W_t := (H_t + \lambda I)^{-1}, \ H_t := \nabla^2 F(\theta_{t-1}) . \tag{4}$$

For sufficiently smooth convex objectives $F$, Newton's method is known to converge at a quadratic rate: $T = O(\log \log(1/\varepsilon))$ iterations are sufficient to approach the minimum within error $\varepsilon$, as opposed to $T = O(\log(1/\varepsilon))$ for gradient descent and its accelerated variants; see (Boyd & Vandenberghe, 2004, Chapter 9). A fast convergence rate is highly desirable in our setting, as the number of iterations corresponds directly to communication/worker deployment rounds.

In our scheme, the server effectively runs the Newton method with an approximate inverse Hessian:

$$\theta_t = \theta_{t-1} - \alpha_t \bar{W}_t g_t, \tag{5}$$

where $\bar{W}$ is obtained by averaging $q$ independent local estimates produced by the workers:[1]

$$\bar{W} := \frac{1}{q} \sum_{k=1}^{q} \hat{W}_t^{(k)}. \tag{6}$$

**Debiased inverse Hessian sketching:**    We wish to avoid directly inverting the full $d \times d$ Hessian, which practically (when done exactly) costs $O(d^3)$. To this end, we have every worker compute, in parallel, a cheap but low-bias estimate of the inverse Hessian by sketching. Specifically, at every deployment round, every worker independently samples a random sketching matrix $S \in \mathbb{R}^{m \times d}$ (where $m < d$), and approximates the inverse Hessian by its sketched version

$$\hat{W} = S^\top (SHS^\top + \tilde{\lambda}I)^{-1}S, \tag{7}$$

where $\tilde{\lambda} > 0$ is a *modified* regularization parameter. Sketches of the type in (7) have been studied before in the literature (also in the context of high-dimensional regression and optimization). Variations have appeared under various names: "dual sketching" (Zhang et al., 2013), "sketch-and-project" (Gower & Richtárik, 2015), "right-sketch" (Murray et al., 2023), and "feature sketching" (Patil & LeJeune, 2023). Note that the cost of forming $\hat{W}g$ is $O(m^3 + M)$, where $M$ is the cost of multiplying $SHS^\top$. In this paper, we exclusively consider the case where $S$ is a random dense matrix with i.i.d. entries.[2] In this case $M = d^2 m$, which dominates the overall per-iteration cost. Naturally, we are particularly interested in settings where $m$ can be chosen very small compared to $d$, so that cost, $O(d^2 m)$, is substantially smaller than $O(d^3)$.

---

[1]In practice, the workers send the vector $\hat{W}^{(k)} g_t \in \mathbb{R}^d$ to the server, rather the whole matrix $\hat{W}^{(k)} \in \mathbb{R}^{d \times d}$.

[2]By "dense", we mean that $S$ has $\Omega(md)$ many non-zero entries typically, though it's not necessary that all the entries, or even a majority of them, are non-zero.

**Remark 1.1.** *In this paper we do not assume any particular form for $H$ (i.e., that it is readily factorizable $H = X^\top X$, $X \in \mathbb{R}^{N \times d}$, a setting which is quite common in the sketching and distributed optimization literature; for example (Pilanci & Wainwright, 2015)). In fact, sketched measurements $SHS^\top$ may be obtainable even in settings when $F$ is not available in analytic form, but only through a program (oracle) that calculates it - for example by automatic differentiation (Paszke et al., 2017).*

A key question is how to choose $m$ (and in accordance $\tilde{\lambda}$). Define the $\lambda$- effective dimension of $H$,[3]

$$\mathsf{d}_H(\lambda) := \mathrm{tr}(H(H + \lambda I)^{-1}). \tag{8}$$

As we shall see in Section 2, results from asymptotic random matrix theory imply that, when $S$ is an i.i.d. dense sketching matrix and **provided** that $m > \mathsf{d}_H(\lambda)$, the choice

$$\tilde{\lambda} := \lambda \left(1 - \frac{1}{m}\mathsf{d}_H(\lambda)\right) \tag{9}$$

results in $\hat{W}$ which is a *low-bias* estimator of the true inverse Hessian $W$. Note that exactly computing the effective dimension is resource-intensive, and essentially requires inverting the Hessian—the very operation we wanted to avoid. A key component of our proposed scheme is a novel low-complexity procedure, based on ideas from asymptotic random matrix theory, for adaptively selecting $m = O(\mathsf{d}_H(\lambda))$ and $\tilde{\lambda}$, by only observing sketched measurements of $H$.

**Remark 1.2.** *When $H$ is effectively low-rank, that is, it exhibits fast spectral decay, $\mathsf{d}_H(\lambda)$ may be much smaller than $d$. For example, when the spectrum has power decay $\lambda_k(H) \propto k^{-\alpha}$ and $\lambda = O(1)$, $\mathsf{d}_H(\lambda) = O(d^{1-\alpha})$ when $\alpha < 1$, $\mathsf{d}_H(\lambda) = O(\log d)$ when $\alpha = 1$ and $\mathsf{d}_H(\lambda) = O(1)$ when $\alpha > 1$. Hessian spectral with power decay are common in regression problems involving kernels (Wainwright, 2019; Yang et al., 2017).*

**Outline and Contributions:**

- In Section 2 we review fundamental results from asymptotic random matrix theory (RMT), specifically the Marchenko-Pastur law and its formulation using deterministic equivalents, and demonstrate how they lead naturally to an asymptotically unbiased inverse Hessian estimator, from i.i.d. sketching matrices $S$. The connection to the Marchenko-Pastur law treats in a uniform framework—and recovers almost immediately—earlier results on Hessian sketch debiasing for distributed Newton methods (Derezinski et al., 2020; Zhang & Pilanci, 2023).

- In Section 3 we propose a *novel data-driven method* for selecting $m$ and $\tilde{\lambda}$ using only the sketched Hessian, avoiding the need to compute the effective dimension exactly. The approach is a natural outgrowth of the RMT framework described previously, and is fully adaptive: no tuning parameters are needed. Section 3.1 further provides non-asymptotic error guarantees for our method, focusing on the case where $S$ is an i.i.d. Gaussian matrix. Notably, the error bounds obtained in this setting do not depend on the Hessian's condition number, nor require $m$ to be proportional to $d$ in any particular way (in RMT, such bounds are sometimes called "dimension-free").

- In Section 4 we derive convergence guarantees for the Newton method with inexact Hessians; to wit, when the exact inverse Hessian $W$ in (2)-(4) is replaced by an $\eta$-accurate estimate $\bar{W}$. By ensuring a sufficiently small $\eta$, the algorithm achieves an arbitrarily fast linear convergence rate. Our results are proved for self-concordant functions $F(\cdot)$, and to our knowledge are novel and may be of independent interest. Combined with the results of the previous section, we obtain a non-asymptotic, end-to-end convergence guarantee for the entire parallel method (with Gaussian sketches), which remarkably does not depend on the condition number of the Hessians.

- Lastly, Section 5 is dedicated to experiments, on both synthetic and real-world data.

## 1.1 RELATED WORKS

Distributed second-order optimization methods have been widely studied for large-scale machine learning. Notable examples include DANE (Shamir et al., 2013), which addresses communication efficiency, AIDE (Reddi et al., 2016), designed for accelerated convergence, and DiSCO (Zhang & Lin, 2015), which focuses on distributed second-order methods for convex optimization problems.

---

[3]$\mathsf{d}_H(\lambda)$ is also called the "effective degrees of freedom" in ridge regression (Hastie et al., 2009), when the design matrix $X \in \mathbb{R}^{N \times d}$ is $H = X^\top X$.

The most closely related body of literature to our work centers around the averaging of inexact Newton steps, which has proven effective when combined with randomized linear algebra. GIANT (Wang et al., 2017) uses the average of local Newton steps as the global step and Determinantal Averaging (Derezinski et al., 2020) provides an unbiased averaging technique for distributed Newton methods. Surrogate sketching (Derezinski et al., 2020) was introduced for distributed Newton's method, revealing improvements via a simple shrinkage strategy. The work (Zhang & Pilanci, 2023) introduced an optimal debiasing method for distributed Newton's method when the effective dimension of the Hessian is known. We remark that the setting being addressed in distributed optimization is typically different than ours: in the former, the objective (which is often the empirical loss over a data set, i.e. in federated learning (Kairouz et al., 2021)) is spread over different machines, with the goal of minimizing the overall transfer of data between machines. In constrast, in our setting all machines have access to the full data, and the goal is to leverage parallelism to reduce the overall runtime and number of deployment rounds. The literature on distributed optimization, and distributed second-order methods in particular, is vast and growing; additional recent references include (Safaryan et al., 2022; Agafonov et al., 2022; Qian et al., 2022; Elgabli et al., 2022; Bischoff et al., 2021), among others. Lastly, there exists a large related corpus of works in optimization involving Hessian subsampling or sketching, for cases where the Hessian admits a good low-rank approximation, though not necessarily in a distributed setting. Examples include (Roosta-Khorasani & Mahoney, 2019; Gower et al., 2019; Frangella et al., 2022), among others. The basic idea of approximating the Hessian in second-order optimization (quasi-Newton methods) is essentially classical, cf. the book (Nocedal & Wright, 1999). Additional recent references exploring related themes include (Gupta et al., 2018; Vyas et al., 2024; Jahani & Rusakov, 2022; Doikov et al., 2023; Na et al., 2022), among many others.

## 2    DEBIASING FROM I.I.D SKETCHES AND RANDOM MATRIX THEORY

This section concerns inverse sketched Hessian debiasing for sketches $S \in \mathbb{R}^{m \times d}$ which have i.i.d. entries with mean zero $\mathbb{E}[S_{ij}] = 0$. We normalize the variance as $\mathbb{E}[S_{ij}]^2 = 1/m$; this normalization is such that $\mathbb{E}[S^\top S] = I$, equivalently, $S : \mathbb{R}^d \to \mathbb{R}^m$ is an isometry in expectation.

Let $H \succeq 0$ be a positive definite matrix (e.g., the full Hessian at the current iteration). The spectral properties of the matrix $SHS^\top \in \mathbb{R}^{m \times m}$, which appears in the sketch (7), are of utmost importance for what follows. Note that this matrix has the same non-zero eigenvalues[4] as the matrix $H^{1/2}S^\top SH^{1/2}$: the latter is a *sample covariance* matrix (making the analogy explicit: $m$ is the number of samples, $H$ is the population covariance, and $x_i := \sqrt{m}H^{1/2}r_i$ are independent "samples", $r_1, \ldots, r_m$ being the rows of $S$); accordingly, $SHS^\top$ is sometimes called the companion sample covariance. Sample covariance matrices, and their spectral properties, have been a central object of investigation in random matrix theory (RMT); cf. (Bai & Silverstein, 2010). A foundational result is the Marchenko-Pastur law (Marčenko & Pastur, 1967), which describes the high-dimensional global behavior of the eigenvalues of $H^{1/2}S^\top SH^{1/2}$ (limiting spectral distribution).

**The Marchenko-Pastur equation:**    For[5] $z \in \mathbb{C}^+$ let $\mathsf{s}(z) \in \mathbb{C}^+$ be the unique solution of

$$\frac{1}{\mathsf{s}(z)} = -z + \frac{1}{m} \operatorname{tr} \left( H(I + \mathsf{s}(z)H)^{-1} \right) \tag{10}$$

subject to the constraint $\mathsf{s}(z) \in \mathbb{C}^+$. It is known that (see (Bai & Silverstein, 2010)) $\mathsf{s}(z)$ is the Stieltjes transform[6] of a *compactly supported probability measure*, whose support lies in $\mathbb{R}_{\geq 0}$; this measure is the **(companion) Marchenko-Pastur law** associated with the population covariance $H$ (and depends only on its eigenvalues) and the number of samples $m$. Moreover: 1) The function $z \mapsto \mathsf{s}(z)$ extends continuously to $\mathbb{R}$, except possibly at $z = 0$; 2) for $x \in \mathbb{R}$ outside the support, the function $x \mapsto \mathsf{s}(x)$ is increasing; in particular, for negative $x < 0$, it is positive and increasing. In this paper, we shall restrict our attention to only negative real valued $z$ in (10), understanding that it

---

[4]This is a standard consequence of the elementary determinant identity $\det(I + AB) = \det(I + BA)$.

[5]Here $\mathbb{C}^+$ is the complex upper halfplane, $\Im(z) > 0$.

[6]For a finite measure $\mu$ on $\mathbb{R}$, its Stieltjes transform is the function $\mathsf{s}_\mu : \mathbb{C}^+ \to \mathbb{C}^+$ given by $\mathsf{s}_\mu(z) = \int \frac{1}{x-z} d\mu(x)$. Using the Stieltjes inversion formula, the measure $\mu$ can be recovered from $\mathsf{s}_\mu$ in the following sense: for every $\varphi(\cdot)$ bounded and continuous, $\int \varphi(t)d\mu(t) = \lim_{\eta \to 0+} \frac{1}{\pi} \int \varphi(x) \Im \mathsf{s}_\mu(x + i\eta)dx$.

holds also for $z < 0$ real. For more details, cf. (Silverstein & Choi, 1995; Bai & Silverstein, 2010). Finally, note that the function $z \mapsto \mathsf{s}(z)$ is invertible, the inverse function given by

$$\Psi(\mathsf{s}) = \frac{1}{m} \operatorname{tr}(H(I + \mathsf{s}H)^{-1}) - \frac{1}{\mathsf{s}}. \tag{11}$$

The Marchenko-Pastur equation (10) may equivalently be written as $z = \Psi(\mathsf{s}(z))$.

In RMT, it is typical to consider the *proportional growth* asymptotic regime: $m, d \to \infty, m = \Theta(d)$.

**Definition 1.** *For (small) constant $\tau > 0$, assumption set [Asymp($\tau, H, \xi$)] holds if: (1) bounded aspect ratio (proportional growth): $\tau < \frac{m}{d} < 1/\tau$; (2) bounded operator norm: $\|H\| < 1/\tau$; (3) $H$ is invertible; (4) $H$ has non-degenerate spectrum: at least $\tau d$ eigenvalues are larger than $\tau$.(5) $\sqrt{m}S_{ij} \sim \xi$, with their law $\xi$, satisfying: $\mathbb{E}[\xi] = 0, \mathbb{E}[\xi^2] = 1$, and $\mathbb{E}[|\xi|^k] < \infty$ for all $k > 0$.*

Denote by $\hat{\mathsf{s}}(z)$ the Stieltjes transform of the empirical eigenvalue distribution of $SHS^\top$:

$$\hat{\mathsf{s}}(z) = \frac{1}{m} \operatorname{tr}(SHS^\top - zI)^{-1} = \frac{1}{m} \sum_{i=1}^{m} \frac{1}{\lambda_i(SHS^\top) - z}. \tag{12}$$

Note that $\hat{\mathsf{s}}(z)$ is a random function, since $S$ is random. The following is a (quantitative) statement of the Marchenko-Pastur law (Knowles & Yin, 2017); for simplicity, we focus on the case $z < 0$.

**Theorem 2.1.** *Set $\eta, D > 0$. Assume [Asymp($\tau, H, \xi$)]. W.p. $1 - O(m^{-D})$, simultaneously for all $-1/\tau \leq z \leq -\tau$,*

$$|\hat{\mathsf{s}}(z) - \mathsf{s}(z)| = O\left(m^{-1/2+\eta}\right). \tag{13}$$

*The constants in the $O(\cdot)$ notation depend on $\tau, \eta, D$.*

*Proof.* See Theorem 3.7 and Remark 3.10 in (Knowles & Yin, 2017). □

**Deterministic equivalent for the sample covariance:** The Marchenko-Pastur law (Theorem 2.1) describes the eigenvalue distribution of $SHS^\top$, equivalently $H^{1/2}S^\top SH^{1/2}$, and relates it to that of $H$ (the connection made via the Marchenko-Pastur equation (10)). One can show something stronger: the entire (random) resolvent $(H^{1/2}S^\top SH^{1/2} - zI)^{-1}$ is in fact close (entrywise, *not* in operator norm) to the resolvent of $H$ (which is deterministic), with appropriate shifting and shrinkage. A result of the following kind is sometimes referred to in the RMT literature as a "deterministic equivalent", cf. (Couillet & Debbah, 2011; Bai & Silverstein, 2010).

**Theorem 2.2.** *Set $\eta, D > 0$. Assume [Asymp($\tau, H, \xi$)]. For any $u, v \in \mathbb{R}^d, \|u\|, \|v\| = 1$, the following holds. W.p. $1 - O(m^{-D})$, simultaneously for all $-1/\tau^{-1} \leq z \leq -\tau$,*

$$u^\top (H^{1/2}S^\top SH^{1/2} - zI)^{-1}v = u^\top(-z\mathsf{s}(z)H - zI)^{-1}v + O(m^{-1/2+\eta}). \tag{14}$$

*Above, the constants in the $O(\cdot)$ notation depend on $\tau, \eta, D$ and are uniform in $u, v$. In particular,*

$$\left\| \mathbb{E}\left[(H^{1/2}S^\top SH^{1/2} - zI)^{-1}\right] - (-z\mathsf{s}(z)H - zI)^{-1} \right\| = O(m^{-1/2+\eta}) \tag{15}$$

*Proof.* See Eq. (3.3) and Theorem 3.7 in (Knowles & Yin, 2017). □

**Deterministic equivalent for the sketched Hessian:** Recently, a deterministic equivalent for the sketched matrix (7) was obtained by (LeJeune et al., 2022). We cite a quantitative version of their result (LeJeune et al., 2022, Theorem 4.1).

**Theorem 2.3.** *Set $\eta, D > 0$. Assume [Asymp($\tau, H, \xi$)], and furthermore $\lambda_{\operatorname{rank}(H)}(H) \geq \tau$. For any $u, v \in \mathbb{R}^d, \|u\|, \|v\| = 1$, w.p. $1 - O(m^{-D})$, simultaneously for all $-1/\tau \leq z \leq -\tau$,*

$$u^\top S^\top (SHS^\top - zI)^{-1}Sv = u^\top (H - (\mathsf{s}(z))^{-1}I)^{-1}v + O(m^{-1/2+\eta}). \tag{16}$$

*In particular,*

$$\left\| \mathbb{E}\left[S^\top (SHS^\top - zI)^{-1}S\right] - (H - (\mathsf{s}(z))^{-1}I)^{-1} \right\| = O(m^{-1/2+\eta}). \tag{17}$$

*Proof.* The result is a consequence of Theorem 2.2; for completeness, let us repeat the argument of (LeJeune et al., 2022), assuming $H$ is full rank (see their paper for a generalization). Denote $\tilde{u} := H^{-1/2}u$, $\tilde{v} := H^{-1/2}v$, so that $\|\tilde{u}\|, \|\tilde{v}\| \le 1/\sqrt{\tau} = O(1)$. We have w.p. $1 - O(m^{-D})$,

$$
\begin{aligned}
u^\top S^\top (SHS^\top - zI)^{-1} Sv &= \tilde{u}^\top H^{1/2} S^\top (SHS^\top - zI)^{-1} SH^{1/2}\tilde{v} \\
&= \tilde{u}^\top (H^{1/2}S^\top SH^{1/2} - zI)^{-1} H^{1/2}S^\top SH^{1/2}\tilde{v} = \tilde{u}^\top \tilde{v} + z\tilde{u}^\top (H^{1/2}S^\top SH^{1/2} - zI)^{-1}\tilde{v} \\
&= \tilde{u}^\top \tilde{v} - \tilde{u}^\top (\mathsf{s}(z)H + I)^{-1}\tilde{v} + O(m^{-1/2+\eta}) = u^\top (H + (\mathsf{s}(z))^{-1}I)^{-1}v + O(m^{-1/2+\eta}).
\end{aligned}
$$

$\square$

## 3 AN ADAPTIVE SKETCHING AND DEBIASING PROCEDURE

Theorem 2.3 suggests a clear path for obtaining a low-bias estimate of the inverse Hessian from the sketch (7), namely: one should choose $\tilde{\lambda} > 0$ such that $\mathsf{s}(-\tilde{\lambda}) = 1/\lambda$, *when such a root exists.*

**Lemma 3.1.** *A solution $\tilde{\lambda} > 0$ satisfying $\mathsf{s}(-\tilde{\lambda}) = 1/\lambda$ exists if and only if $m > \mathsf{d}_H(\lambda)$. When this is the case, $\tilde{\lambda}$ is given by (9).*

*Proof.* By merit of being the Stieltjes transform of a finite measure supported on $\mathbb{R}_{\ge 0}$, $\mathsf{s}(\cdot)$ maps bijectively $(-\infty, 0)$ to $(0, \mathsf{s}(0))$, hence a solution $\tilde{\lambda} > 0$ exists if and only if $\mathsf{s}(0) > 1/\lambda$. Taking the limit $z \to 0+$ in (10), we find (after routine algebraic manipulation) that $\mathsf{s}(0)$ solves $\mathsf{d}_H(1/\mathsf{s}(0)) = m$. Since the function $\mu \mapsto \mathsf{d}_H(\mu)$ is decreasing for positive $\mu$, we deduce that $1/\mathsf{s}(0) < \lambda$ if and only if $m > \mathsf{d}_H(\lambda)$. When this is the case, $\tilde{\lambda} = \Psi(1/\lambda)$ where $\Psi$ is the explicit inverse (11). $\square$

**Remark 3.2.** *Earlier works on distributed Newton's method (Derezinski et al., 2020; Zhang & Pilanci, 2023), considered debiasing for a closely related but different Hessian sketch. Specifically, they assume the Hessian has the form $H = X^\top X$ where $X \in \mathbb{R}^{N \times d}$; such Hessians naturally appear in machine learning problems, for example in the (reguarlized) empirical loss minimization for a generalized linear model (GLM). (Zhang & Pilanci, 2023) considered (among others), the sketch $X^\top S^\top SX$, and calculated the appropriate shrinkage factor for asymptotically debiasing the inverse: $(\gamma X^\top S^\top SX + \lambda I)^{-1} \simeq (H + \lambda I)^{-1}$ where $\gamma := 1/\left(1 - \frac{1}{m}\mathsf{d}_H(\lambda)\right)$, provided that $m > \mathsf{d}_H(\lambda)$, under an asymptotic setting as in Theorem 2.2 (though without making the connection to the Marchenko-Pastur law explicit). In the scheme they considered, the effective dimension was assumed to be known. We remark that Theorem 2.2 holds verbatim if $H^{1/2}$ is replaced by $X^\top$, and the ratio $\tau < d/N < 1/\tau$ is bounded (Knowles & Yin, 2017). Rewritting (15) as $(H + (\mathsf{s}(z))^{-1}I)^{-1} \simeq \left((-z\mathsf{s}(z))^{-1}X^\top S^\top SX + (\mathsf{s}(z))^{-1}I\right)^{-1}$, setting $z = -\tilde{\lambda}$ recovers their proposed shrinkage factor. Accordingly, the adaptive procedure we propose below applies to that sketching scheme as well, with the obvious modifications.*

Note that we do not have access to the function $\mathsf{s}(\cdot)$ directly (to calculate it numerically via (10), one needs to invert the full Hessian)—but we *can* estimate it from the sketched measurements $SHS^\top$, using Theorem 2.1. Accordingly, we can estimate $\tilde{\lambda}$ by the solution $\hat{\lambda}$ of the (random) equation

$$\hat{\mathsf{s}}(-\hat{\lambda}) = 1/\lambda, \quad \hat{\lambda} > 0, \tag{18}$$

if it exists. To have $\hat{\lambda} \approx \tilde{\lambda}$, we need that $m > \mathsf{d}_H(\lambda)$, and we do not know this a priori - therefore we also need a way to select large enough $m$ prior to solving (18). Moreover, to ensure small error in Theorems 2.1-2.3, one needs to work with $z$ bounded away from 0. As $\tilde{\lambda} = \lambda(1 - \frac{1}{m}\mathsf{d}_H(\lambda))$, to ensure that this is $\Omega(\lambda)$ we need to have $m \ge (1 + \Omega(1))\mathsf{d}_H(\lambda)$. Specifically, we shall aim to choose $m$ between $1.5\mathsf{d}_H(\lambda) \le m \le 4\mathsf{d}_H(\lambda)$, the constants chosen somewhat arbitrarily.

**Testing for a good $m$.** We devise a test that, given $m$, with high probability: (a) rejects if $m < 1.5\mathsf{d}_H(\lambda)$; and (b) accepts if $m \ge 2\mathsf{d}_H(\lambda)$. Set $z_0 = -5\lambda/12$. The tests accepts if $\hat{\mathsf{s}}(z_0) > 1/\lambda$, and rejects otherwise.
If $m < 1.5\mathsf{d}_H(\lambda)$ then either there is no solution $\tilde{\lambda} > 0$ to $\mathsf{s}(-\tilde{\lambda}) = 1/\lambda$, or there is one, specifically $\tilde{\lambda} = \lambda(1 - \frac{1}{m}\mathsf{d}_H(\lambda)) \le \lambda/3 < -z_0$. Since $z \mapsto \mathsf{s}(z)$ is positive and increasing for $z < 0$, necessarily $\mathsf{s}(z_0) < \mathsf{s}(-\tilde{\lambda}) = 1/\lambda$. Provided that $m$ is large enough, w.h.p. $\hat{\mathsf{s}}(z_0) < 1/\lambda$ as well,

hence the test rejects.

If $m \geq 2\mathsf{d}_H(\lambda)$ then $\tilde{\lambda} = \lambda(1 - \frac{1}{m}\mathsf{d}_H(\lambda)) \geq \lambda/2 > -z_0$, hence $\mathsf{s}(z_0) > \mathsf{s}(-\tilde{\lambda}) = 1/\lambda$. Provided that $m$ is large enough, w.h.p. $\hat{\mathsf{s}}(z_0) > 1/\lambda$ as well, hence the test accepts.

**Searching for $m$ by doubling.** Start with $m = m_0$ large enough; proceeds iteratively, doubling $m$ after each iteration $m := 2m$. At every iteration, apply the above test, halting if it accepts and continuing if it rejects. W.h.p., (a) we do not halt on any $m < 1.5\mathsf{d}_H(\lambda)$; (b) we halt on the first $m$ satisfying $m \geq 2\mathsf{d}_H(\lambda)$, and therefore return $m$ satisfying $m \leq 4\mathsf{d}_H(\lambda)$.

**Computational complexity.** W.h.p., the doubling procedure stops within $\log_2(4\mathsf{d}_H(\lambda)/m_0)$ iterations. Each iteration $t$ costs at most $O(d^2 m_t)$, $m_t = 2^t m_0$, so the total cost is $O(d^2 \min\{\mathsf{d}_H(\lambda), m_0\})$ and dominated by the cost of multiplying $SHS^\top$ for $S \in \mathbb{R}^{m_t \times d}$ at the last iteration.

Having found large enough $m$, we can solve the equation (18) by binary search, and form the sketch (7) with $\hat{\lambda}$ in place of $\tilde{\lambda}$. For the reader's convenience, the overall adaptive sketching procedure—which is performed (in parallel) by every worker—is summarized in Algorithms 1-2.

---

**Algorithm 1**

---

1: **procedure** *Choose-Sketching-Dimension-IID*($H, \lambda, m_0$)
2:     $m = m_0$
3:     **while** $m < d$ **do**
4:         Sample i.i.d. sketch $S \in \mathbb{R}^{m \times d}$ and form $SHS^\top$.
5:         Compute eigenvalues $\lambda_1(SHS^\top) \geq \ldots \geq \lambda_m(SHS^\top)$.
6:         **if** $\hat{\mathsf{s}}(-5\lambda/12) > 1/\lambda$ **then**                    $\triangleright$ $\hat{\mathsf{s}}(z)$ defined in (12).
7:             **return** m                    $\triangleright$ Guarantee: w.h.p., $1.5\mathsf{d}_H(\lambda) \leq m \leq 4\mathsf{d}_H(\lambda)$.
8:         **else**
9:             m := 2m
10:        **end if**
11:    **end while**
12:    **return** m
13: **end procedure**

---

**Algorithm 2**

---

1: **procedure** *Estimate-Inverse-Hessian-IID* ($H, \lambda, m$)
2:     Sample sketch $S \in \mathbb{R}^{m \times d}$ and form $SHS^\top$.
3:     Compute eigenvalues $\lambda_1(SHS^\top) \geq \ldots \lambda_m(SHS^\top)$.
4:     **if** $\hat{\mathsf{s}}_m(0) \leq 1/\lambda$ **then**
5:         ERROR; set $\hat{\lambda} = 5\lambda/12$.
6:     **else**
7:         Find $5\lambda/12 \leq \hat{\lambda} \leq \lambda$ such that $\hat{\mathsf{s}}(-\hat{\lambda}) = 1/\lambda$.                    $\triangleright$ Can be done by binary search.
8:     **end if**
9:     **return** $(\hat{\lambda}, \hat{W})$ where $\hat{W} = S^\top(SHS^\top + \hat{\lambda}I)^{-1}S$                    $\triangleright$ (In practice, return $\hat{W}g$ where $g \in \mathbb{R}^d$ is the current gradient.)
10: **end procedure**

---

It is straightforward to establish the asymptotic validity of our adaptive sketching procedure.

**Theorem 3.3.** *Set $\eta, D > 0$, let $m_0$ be an initial sketch size. Assume [**Asymp**($\tau, H, m_0, \xi$)] (in particular $m_0 = \Omega(d)$), and furthermore that $\lambda_{\mathrm{rank}(H)}(H) \geq \tau$. Let $\hat{m}$ be the output of Algorithm 1, and $(\hat{\lambda}, \hat{W})$ be the output of Algorithm 2 (the latter given input $m = \hat{m}$). Then w.p. $1 - O(m_0^{-D})$:*

- $\max\{m_0, 1.5\mathsf{d}_H(\lambda)\} \leq \hat{m} \leq \max\{m_0, 4\mathsf{d}_H(\lambda)\}$.
- $|\hat{\lambda} - \tilde{\lambda}| = O(m_0^{-1/2+\eta})$.
- $\|\mathbb{E}[\hat{W}] - W\| = O(m_0^{-1/2+\eta})$.

*The constants in the $O(\cdot)$ notation may depend on $\eta, D > 0$ and also $\lambda > 0$.*

*Proof.* This is a straightforward consequence of Theorems 2.1-2.3. We omit the details.    $\square$

### 3.1 Non-Asymptotic Guarantees for Gaussian Sketches

The guarantees provided by Theorem 3.3 are asymptotic, in that they do not depend explicitly on crucial problem parameters (such as $\lambda, \|H\|$); in particular, we implicitly assume that $H$ is well-conditioned. When $S \in \mathbb{R}^{m \times d}$ is an i.i.d. Gaussian matrix, $S_{ij} \sim \mathcal{N}(0, 1/m)$, we are able to provide **non-asymptotic** error guarantees for our adaptive sketching algorithm. Our bounds are dimension-free, and remarkably do not depend explicitly on the condition number of $H$. The theorems below are the main technical part of this paper; their proofs are deferred to the appendix.

Theorems 3.4 and 3.5 respectively provide non-asymptotic guarantees for Algorithms 1 and 2:

**Theorem 3.4.** *Assume that $S$ has i.i.d. Gaussian entries. Let $\delta \in (0, 1)$. Suppose that $m_0 \geq C(1 + \log(1/\delta))$ for large enough constant $C$. W.p. $1 - \delta$, Algorithm 1 outputs $\hat{m}$ such that*

$$\max\{1.5 \mathsf{d}_H(\lambda), m_0\} \leq \hat{m} \leq \max\{4 \mathsf{d}_H(\lambda), m_0\}. \tag{19}$$

**Theorem 3.5.** *Assume that $S$ has i.i.d. Gaussian entries. Let $\delta \in (0, 1)$ and $\varepsilon > 0$. Suppose Algorithm 2 is run with $m \geq 1.5 \mathsf{d}_H(\lambda)$ and that moreover $m \geq C \frac{1 + \log(1/\delta)}{\varepsilon^2}$ for large enough constant $C$. W.p. $1 - \delta$, the output $\hat{\lambda}$ satisfies $|\hat{\lambda} - \tilde{\lambda}| \leq \varepsilon \tilde{\lambda}$, where $\tilde{\lambda} = \lambda \left(1 - \frac{1}{m} \mathsf{d}_H(\lambda)\right)$.*

Our final non-asymptotic result concerns the concentration of the aggregate inverse Hessian estimate used by the server. To wit, fix $m \geq 1.5 \mathsf{d}_H(\lambda)$, and let $\hat{W}^{(1)}, \ldots, \hat{W}^{(q)}$ be the outputs of $q$ independent runs of Algorithm 2, representing $q$ workers producing estimates of the Newton step in parallel. Denote their average $\bar{W} = \frac{1}{q} \sum_{k=1}^{q} \hat{W}^{(k)}$, and the error matrix:

$$\bar{\mathcal{E}} := (H + \lambda I)^{1/2} \bar{W} (H + \lambda I)^{1/2} - I. \tag{20}$$

Later on, we will obtain error bounds for the quasi Newton method in terms of $\|\bar{\mathcal{E}}\|$.

**Theorem 3.6.** *Assume that $S$ has i.i.d. Gaussian entries. Suppose that $m \geq 1.5 \mathsf{d}_H(\lambda)$, and let $q \geq 1$, $\delta \in (0, 1)$ satisfy $1/\delta \leq \exp(O(d))$, $q \leq \exp(O(d))$. Let $\exp(-O(d)) \leq \varepsilon \leq O(1)$ be small enough. If moreover*

$$m \geq C \frac{1}{\varepsilon^2}, \quad q \geq C \frac{d \log d}{m} \frac{\log(1/\delta)}{\varepsilon^2} \tag{21}$$

*for large enough $C > 0$, then w.p. $1 - \delta$, $\|\bar{\mathcal{E}}\| \leq \varepsilon$.*

**Remark 3.7.** *Up to the logarithmic factor (which we believe is an artifact), the dimensional dependence of $q$ in (21) is optimal. To see this, suppose that $H = 0, \lambda = 1$, so $\hat{W}^{(k)} = S^{(k)\top} S^{(k)}$. In particular $\mathrm{rank}(\bar{W}) = qm$, therefore $\|\bar{\mathcal{E}} - I\| \geq 1$ when $qm < d$. In fact, $\bar{\mathcal{E}}$ is a Wishart (Gaussian sample covariance) matrix corresponding to $mq$ samples, therefore by standard estimates $\mathbb{E}\|\bar{\mathcal{E}} - I\| \asymp \frac{d}{mq} \vee \sqrt{\frac{d}{mq}}$, hence the dependence on $\varepsilon$ is correct as well.*

## 4 Convergence Results for the Approximate Newton Method

In this section we present guarantees for the Newton method with approximate Hessian estimates. These bounds are relevant beyond the parallel scheme considered in this paper, and accordingly are stated in general terms. Let $G : \mathbb{R}^d \to \bar{\mathbb{R}}$ be convex and twice-differentiable.[7] We consider generally the Newton-type iteration,

$$\theta_t = \theta_{t-1} - \alpha_t \bar{W}_t g_t, \tag{22}$$
$$g_t := \nabla G(\theta_{t-1}), \tag{23}$$

where $\bar{W}_t$ approximates the inverse Hessian, $\mathsf{H}_t := \nabla^2 G(\theta_{t-1})$, $\bar{W}_t \approx \mathsf{H}_t^{-1}$ and $\alpha_t$ is a step size.

**Definition 2.** *We say (22)-(23) implement an $\eta$-accurate Newton method if the following holds. Define the error matrix, which measures how well $\bar{W}_t$ approximates the true inverse Hessian $\mathsf{H}_t^{-1}$:*

$$\bar{\mathcal{E}}_t := \mathsf{H}_t^{1/2} \bar{W}_t \mathsf{H}_t^{1/2} - I. \tag{24}$$

*Then $\|\bar{\mathcal{E}}_t\| \leq \eta$ for every iteration $t$.*

---

[7] We denote $\bar{\mathbb{R}} = \mathbb{R} \cup \{\infty\}$. With this, the domain of $G$ is $\mathrm{Dom}(G) := \{x : F(x) < \infty\}$.

For a desired precision $\varepsilon > 0$, namely when one desires $\hat{\theta}$ with $G(\hat{\theta}) - G(\theta^\star) \leq \varepsilon$, $\theta^\star := \arg\min G(\theta)$, let $\mathsf{T}(\varepsilon)$ be the smallest $t \geq 1$ (a priori, if it exists) such that $G(\theta_t) - G(\theta^\star) \leq \varepsilon$ for all $t \geq \mathsf{T}(\varepsilon)$. This notation suppresses the dependence on $G$ and the sequence of step sizes $\alpha_t$. We consider a variation where the step size is chosen by backtracking line search, see Algorithm 3.

---

**Algorithm 3** (Backtracking Line Search)

---

1: **procedure** *LineSearch* $(\theta_{t-1}, g_t, \bar{W}_t; G(\cdot))$
2:     **Parameters:** $a, b \in (0, 1)$ line search parameters.
3:     $\alpha = 1$
4:     **while** $G(\theta_{t-1} - \mu \bar{W} g_t) > G(\theta_{t-1}) - a g_t^\top (\alpha \bar{W} g_t)$ **do**
5:         $\alpha := b\alpha$
6:     **end while**
7:     **return** $\alpha$
8: **end procedure**

---

This paper focuses on the case when $G$ is self-concordant, which is a standard setting in the literature on second order convex optimization, cf. (Boyd & Vandenberghe, 2004). We remark that previous works, for example (Wang et al., 2018; Dereziński & Mahoney, 2019), derived convergence guarantees for the approximate Newton method under different assumptions, namely when the Hessian of $G$ is Lipschitz. Due to space constraint, we do not present their results here, though they are applicable for our setting as well. For self-concordant objectives, we are not aware of similar results previously being written down in the literature, hence our results may be of independent interest.

**Definition 3.** *We say a univariate convex function $G : \mathbb{R} \to \bar{\mathbb{R}}$ is self-concordant if it is thrice-differentiable and $|G'''(x)| \leq 2G''(x)^{3/2}$ for any $x$ in its domain. A multivariate convex function $G : \mathbb{R}^d \to \mathbb{R}$ is self-concordant if any 1-dimensional restriction $G_{x,y}(t) = G(x + ty)$ is self-concordant.*

When $f$ is self-concordant, the objective $G(\theta) = f(\theta) + \frac{\lambda}{2}\|\theta\|^2$ in (1) is self-concordant as well, as the sum of self-concordant function is itself self-concordant, cf. (Boyd & Vandenberghe, 2004).

**Theorem 4.1.** *Suppose that (a) $G$ is self-concordant; (b) the Newton method is $\eta$-accurate with $\eta < 1/5$; and (c) $\alpha_t$ is chosen by backtracking line search. For large enough numerical $C > 0$, let*

$$T_0 = C\frac{G(\theta_0) - G(\theta^\star)}{ab}, \quad T_1(\varepsilon) = C \log\log(1/\varepsilon), \quad T_2(\varepsilon) = C\frac{\log(\eta/\varepsilon)}{\log(1/\eta)},$$

*where $a, b$ are the parameters used in backtracking line search (Algorithm 3). Then*

$$\mathsf{T}(\varepsilon) \leq \begin{cases} T_0 + T_1(\varepsilon) & \text{if} \quad \varepsilon \geq \eta \\ T_0 + T_1(\eta) + T_2(\varepsilon) & \text{if} \quad \varepsilon < \eta \end{cases}. \tag{25}$$

The proof of Theorem 4.1 builds on and extends an argument of (Pilanci & Wainwright, 2015). Interestingly, Theorem 4.1 suggests that the approximate Newton method exhibits two stages: (1) While the error is above the noise floor $\varepsilon \geq \eta$, the error decays quadratically fast. (2) Upon reaching the noise floor, $\varepsilon \leq \eta$, the convergence rate becomes linear; however, the linear rate is proportional to $\eta$, that is the error decays as $\sim (O(\eta))^t$. When $\eta$ is very small this decay rate is very fast.

Finally, Theorem 4.1 may be readily combined with Theorem 3.6 to yield error bounds for the entire end-to-end parallel method: under the setting of Theorem 3.6, w.h.p. (and up to $\mathrm{polylog}(d)$ factors),

$$\eta = \tilde{O}\left(\frac{1}{\sqrt{m}} + \sqrt{\frac{d}{mq}}\right).$$

With this, let us roughly bound the runtime of our method with parallel workers. We focus on total flops, ignoring that in practice, deployment rounds are typically more expensive than computation. Every round, each worker performs, in parallel, $O(d^2 m)$ flops, where $m = O(\mathsf{d}_H(\lambda))$ is assumed for simplicity to be roughly constant throughout. We bound the number of deployment rounds by $\mathsf{T}(\varepsilon) = O(\log(1/\varepsilon)/\log(1/\eta))$, assuming that $\varepsilon$ is much smaller than $\eta$. Consider two regimes:

- *Massive parallelization*: When $q \gtrsim d$, the Hessian approximation error is $\eta = O(1/\sqrt{m})$ (bias dominated), and the total runtime is $O(d^2 m \log(1/\varepsilon)/\log m)$. Note that this quantity decreases with $m$—hence it is always preferable to pick $m$ to be the smallest possible (which is

| $\alpha$ | $\mathsf{d}_H(\lambda)$ | $m_0$ | Success rate | Avg. dim. |
|---|---|---|---|---|
| 1 | 8.8 | 10 | 1.0 | 20 |
| 2/3 | 59.7 | 10 | 1.0 | 160 |
| 1/2 | 190.4 | 10 | 1.0 | 640 |

Table 1: Success rate of Algorithm 1.

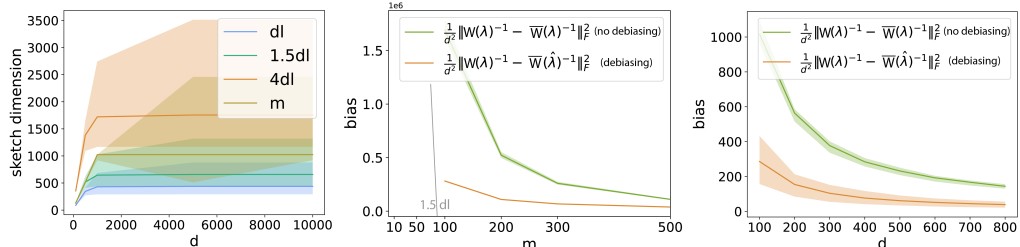

Figure 1: The bias proxy of the bias-corrected inverse Hessian estimator is substantially lower than without bias correction. Rightmost plot: ensemble (R); left and middle plots: ensemble (L). Leftmost plot: the sketching dimension found by Algorithm 1 on ensemble (L). Shaded area: 20%-80% confidence interval; We take $T = 10$ Monte-Carlo trials.

what we do essentially). Note when $m \approx \mathsf{d}_H(\lambda) \ll d$, this runtime substantially improves on $O(d^3 \log \log(1/\varepsilon))$, the runtime of the exact Newton method (with no parallelism). Importantly, this bound does not depend on the conditioning of the objective, unlike in first order methods.

- *Moderate parallelization*: when $d/m \lesssim q \ll d$, the Hessian approximation error is $\eta = O(\sqrt{d/mq})$ (variance dominated), and the total runtime is $O(d^2 m \log(1/\varepsilon)/\log(mq/d))$. At the extreme end of this range, when $q \approx d/m$, let us crudely approximate $\log(mq/d) \approx mq/d - 1$, so that $\log(mq/d) \gtrsim mq/d - 1 \gtrsim mq/d$. Thus, the runtime is roughly $O(d^2 m \log(1/\varepsilon) \log(mq/d)) = O(d^2 m \log(1/\varepsilon)/(mq/d)) = O(d^3 \log(1/\varepsilon)/q)$. For a moderate number of workers $q$ and realistic precision $\varepsilon$, this improves substantially upon the runtime of exact Newton, $O(d^3 \log \log(1/\varepsilon))$. We again stress that this bound does not depend on the condition number of the objective, unlike for first order methods.

## 5 NUMERICAL EXPERIMENTS

We demonstrate the utility and validity of our results via numerical experiments. Due to space contraints, we defer some details and additional experiments to the appendix.

Our first set of experiments aims to demonstrate our adaptive sketching procedure (Algorithms 1-2) independently of any specific optimization context. For the first experiment, we consider diagonal matrices[8] $H$ with eigenvalues $(h_1, \ldots, h_d)$ exhibiting polynomial spectral decay $h_k = k^{-\alpha}$, where $\alpha > 0$ is a parameter. The dimension $d = 10^4$ is fixed and large, and $\lambda = 1$. The sketching matrix is i.i.d. Gaussian (in the appendix, we present results for two more i.i.d. sketching matrices). We run Algorithm 1, repeating for $T = 20$ Monte-Carlo trials, and report the sketching dimension found, see Table 1. Remarkably, our algorithm consistently succeeds, namely finds a sketching dimension that is within $1.5\mathsf{d}_H(\lambda) \le m \le \max\{m_0, 4\mathsf{d}_H(\lambda)\}$ - per Theorems 3.3 and 3.4.

Next, we demonstrate that Algorithm 2 produces low-bias estimates of $W$. Note that the average Frobenius error $\|\hat{W} - W\|_F^2/d^2$ is a reasonable proxy for the bias of $\hat{W}$, see Theorems 2.2-2.3. In the next experiment, we consider two random ensembles of matrices $H$, which we call here (L) and (R), and exhibits respectively slow and fast spectral decay; we defer the details to the appendix, due to space constraints. Figure 1 plots the bias proxy for $m$ increasing, starting from $m = 1.5\mathsf{d}_H(\lambda)$, for the Gaussian sketch. As $m$ increases, the bias decreases (roughly at rate $1/\sqrt{m}$). We also plot the bias proxy for a naive estimator of $W$, without bias correction ($\tilde{\lambda} = \lambda$); clearly, our algorithm has substantially smaller bias throughout.

---

[8]We choose $H$ to be diagonal w.l.o.g. since the Gaussian sketch is invariant to orthogonal transformations.

Our final set of experiments concerns the performance of the proposed optimization strategy in its entirety, focusing on the benefits of bias reduction. We present experimental results for real-world optimization tasks—for both ridge and logistic regression—using publicly available UCI datasets (Chang & Lin, 2011). Due to space constraints, the details appear in the appendix. We compare our method with an approximate parallel Newton method where Hessians are sketched ($m$ chosen by Algorithm 1), but no debiasing is done; that is, $\tilde{\lambda} = \lambda$. It is evident from our results, summarized in Figure 2, that bias correction accelerates convergence, often substantially.

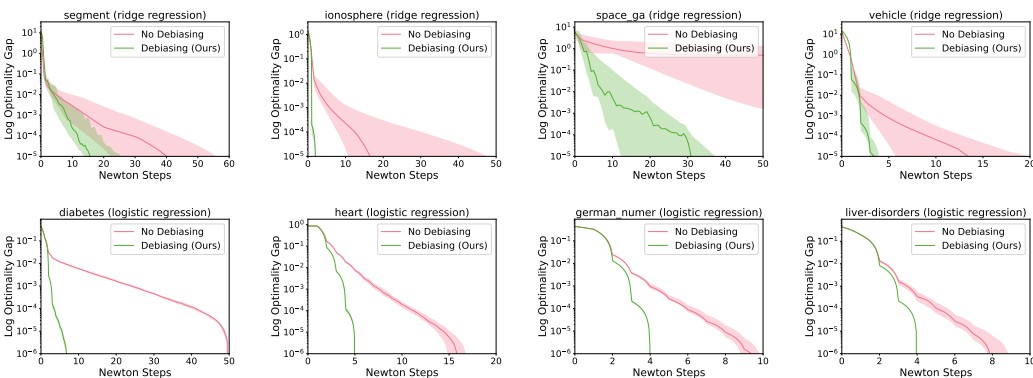

Figure 2: Improved convergence of our parallel sketched Newton method with bias correction. The title of sub-figure corresponds to the dataset used. Repeating for $T = 10$ Monte-Carlo trials, the curve corresponds to the median and the shaded part to a $20\%$-$80\%$ confidence interval.

ACKNOWLEDGMENTS

We are grateful to Daniel Lejeune for inspiring conversations about his work (LeJeune et al., 2022).

Mert Pilanci and Fangzhao Zhang were supported in part by National Science Foundation (NSF) under Grant DMS-2134248; in part by the NSF CAREER Award under Grant CCF-2236829; in part by the U.S. Army Research Office Early Career Award under Grant W911NF-21-1-0242; in part by the Office of Naval Research under Grant N00014-24-1-2164. In addition, Fangzhao Zhang was supported in part by a Stanford Graduate Fellowship.

The work of Elad Romanov was supported in part by the NSF under Grant DMS-1811614 (PI: Donoho), and in part by the personal faculty funds of Prof. David Donoho, for whom he wishes to extend his sincere appreciation.

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

# Appendix

## Table of Contents

# A  ADDITIONAL EXPERIMENTS AND DETAILS

## A.1  EXTENSION OF TABLE 1

We run experiments with random sketching matrices with i.i.d. entries of the following types: (1) Gaussian (G): $S_{i,j} \sim \mathcal{N}(0, 1/m)$; Rademacher (R): $S_{i,j} \sim \mathrm{Unif}(\pm 1/\sqrt{m})$; (3) sparse Rademacher (SR): $S_{i,j} = 0$ w.p. $1 - p$ and $S_{i,j} \sim \mathrm{Unif}(\pm 1/\sqrt{pm})$ w.p. $p$ for $p = 1/10$. The following is an extension of Table 1, to include also the Rademacher (R) and sparse Rademacher (SR) sketch types:

| $\alpha$ | $\mathsf{d}_H(\lambda)$ | $m_0$ | Sketch | Success rate | Avg. dim. |
|---|---|---|---|---|---|
| 1 | 8.8 | 10 | G | 1.0 | 20 |
| 2/3 | 59.7 | 10 | G | 1.0 | 160 |
| 1/2 | 190.4 | 10 | G | 1.0 | 640 |
| 1 | 8.8 | 10 | R | 1.0 | 20 |
| 2/3 | 59.7 | 10 | R | 1.0 | 160 |
| 1/2 | 190.4 | 10 | R | 1.0 | 640 |
| 1 | 8.8 | 10 | SR | 1.0 | 20 |
| 2/3 | 59.7 | 10 | SR | 1.0 | 160 |
| 1/2 | 190.4 | 10 | SR | 1.0 | 640 |

Table 2: Success of Algorithm 1. $T = 20$ Monte-Carlo trials.

Our method appears to be remarkably robust, per the finding in Table 2, to the extent that the table may seem somewhat pointless to include. We nonetheless provide it (and code to run this experiment) as a sort of sanity check.

## A.2  DETAILS FOR FIGURE 1

We provide additional details for the experiments in Figure 1. In the middle and rightmost plots, the estimator $\bar{W}$ is obtained by averaging $q = 500$ local sketched Hessians, both with and without bias correction.

Ensemble (L): We use $L = 10^{-3}$. The matrix $H$ is $H = X^\top X$, where $X \in \mathbb{R}^{n \times d}$ has the singular value decomposition (SVD) $X = UDV^\top$; $U \in \mathbb{R}^{n \times d}, V \in \mathbb{R}^{d \times d}$ have orthonormal columns and $D \in \mathbb{R}^{d \times d}$ is diagonal. $U, V$ are sampled uniformly (Haar) from the corresponding Stieffel manifolds (have a rotationally invariant distribution). For the leftmost plot $n = 10^4$, and $d$ varies. For the middle plot, we fix $d = 10^3$. The diagonal entries of $D$ are random, and satisfy

$$D_{k,k} = (0.9 + \varepsilon_k)^{k/2}, \quad \varepsilon_k \sim \mathcal{N}(0, 10^{-4}).$$

Ensemble (R): We use $\lambda = 10^{-5}$. Similarly to ensemble (L), $H = X^\top X$ for $X \in \mathbb{R}^{n \times d}$, with the singular vectors of $X$ of similarly uniformly random. We use $n = 10^4$, $d = 10^3$. The singular values of $X$ are non-random, with $D_{k,k} = (k/d)^2$.

## A.3  ADDITIONAL SYNTHETIC OPTIMIZATION TASKS

In this section we present additional experiments on synthetic optimization tasks, aiming to test the performance of our proposed end-to-end parallel optimization method. Specifically, we consider ridge and logistic regression. We test both Gaussian and Rademacher sketches (see Section A.1).

**Distribution of covariates (design).**  We generate designs $X \in \mathbb{R}^{n \times d}$ with $n = 10^4$ and $d = 500$. Denote the SVD $X = UDV^\top$. The singular vectors $U \in \mathbb{R}^{n \times d}, V \in \mathbb{R}^{d \times d}$ are uniformly random. The singular values decay exponentially, $D_{k,k} = 0.99^{k/2}$.

**Parameters.**  In all our experiments, $\lambda = 10^{-3}$. For experiments with ridge regression, the number of workers is $q = 10$; for logistic regression, it is $q = 20$.

**Ridge regression.**  The responses are generated according to $y_i = x_i^\top \theta^\star + \mathcal{N}(0, 0.01)$, where the ground-truth regressor $\theta^\star \in \mathbb{R}^d$ is generated according to $\theta^\star \sim \mathcal{N}(0, I)$. The unregularized

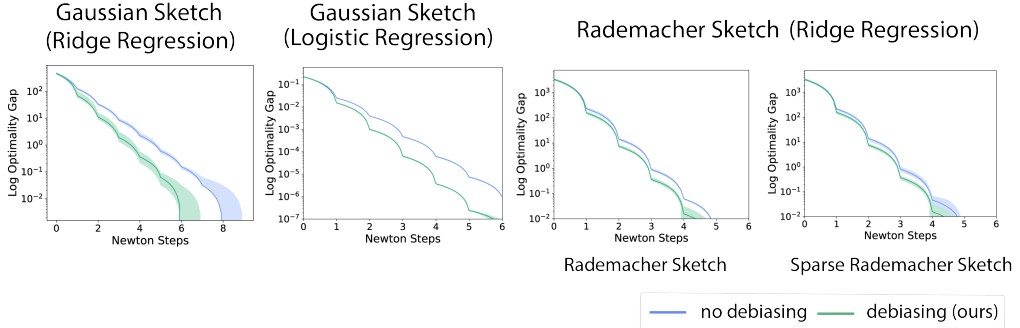

Figure 3: Convergence of the parallel Newton method with Hessian sketching, on synthetic optimization tasks.

objective $F(\theta)$ is the L2 loss:

$$F(\theta) = \frac{1}{n} \sum_{i=1}^{n} (x_i^\top \theta - y_i)^2 = \|X\theta - y\|^2,$$

where $x_1, \ldots, x_n \in \mathbb{R}^d$ are the rows of $X$. Recall that we aim to minimize the regularized objective $G(\theta) := F(\theta) + \frac{\lambda}{2}\|\theta\|^2$.

**Logistic regression.** We generate labels according to

$$y_i = (\operatorname{sign}\left[x_i^\top \theta^\star + \mathcal{N}(0, 10^4)\right] + 1)/2 \in \{0, 1\},$$

where $\theta^\star \sim \mathcal{N}(0, I)$. The function $F(\theta)$ is the log loss:

$$F(\theta) = -\frac{1}{n} \sum_{i=1}^{n} \left[ y_i \log\left(\frac{1}{1 + e^{-x_i^\top \theta}}\right) + (1 - y_i) \log\left(1 - \frac{1}{1 + e^{-x_i^\top \theta}}\right) \right].$$

Figure 3 plots the error $G(\theta_t) - \operatorname{argmin}_\theta G(\theta)$ over the iterations $t$ of the algorithm. Each plot is obtained by $T = 10$ Monte-Carlo trials; the shaded area corresponds to a 20%-80% confidence interval. It is evident that bias correction improves on the the convergence speed of the algorithm over the uncorrected ($\tilde{\lambda} = \lambda$) alternative.

## A.4 DETAILS FOR EXPERIMENTS ON UCI DATA SETS (FIGURE 2)

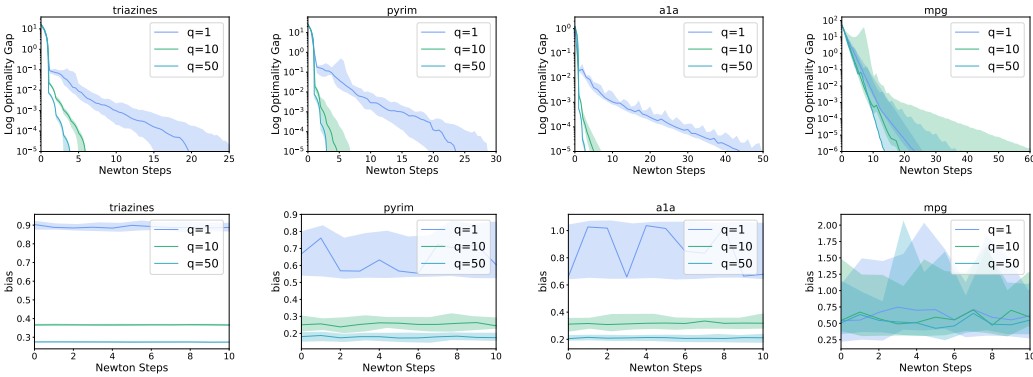

Figure 4: Convergence rate for optimization tasks on additional UCI data sets. Shaded area corresponds to 20%-80% confidence interval.

We provide additional details for the experiments reported in Figure 2, and present results for additional data sets from the UCI data repository (Chang & Lin, 2011).

We run experiments directly on the available data, without additional preprocessing. The objectives, for ridge and logistic regression, are as described in Section A.3, but now the covariates and reponses/labels are given and not generated.

**Parameters.** We use $\lambda = 10^{-3}$ for all experiments. We vary the number of workers between $q \in \{1, 10, 50\}$. For every data set, we run $T = 10$ Monte-Carlo trials.

**Results.** The experiments are reported in Figure 4. The plots on the top correspond to the optimality gap $G(\theta_t) - \operatorname{argmin}_\theta G(\theta)$, $G$ being the regularized objective, as a function of the iteration number $t$. The bottom plots are the inverse Hessian normalized Frobenius error $\|W_t - \bar{W}_t\|_F^2/d^2$. We see that as $q$ grows larger, the algorithm tends to converge faster; though the benefit of further increasing $q$ tends to diminish when $q$ is large.

## A.5 ON THE MERIT OF CENTRALIZED DATA ACCESS VERSUS DISTRIBUTED NEWTON METHODS

### A.5.1 BASELINE REVIEW

Here, we include comparisons with existing distributed Newton's methods, which operate under a slightly different paradigm as they focus on data splitting among workers. These experiments are presented to illustrate that our debiasing approach, which operates under centralized data access within a multi-worker setting, achieves superior acceleration for Newton's methods.

We provide a review on all baseline methods we are comparing against. Consider any loss function $F(x)$ where $x$ is the desired parameter. Given a dataset with $n$ samples, we first split the dataset uniformly across $q$ workers in total, which results in each worker holding $n/q$[9] data points. Then, each worker $i$ ($i \in [q]$), computes the $i$-th component of the Hessian $(\nabla^2 F)_i$ only based on its local data, where the full Hessian is $\sum_{i=1}^n (\nabla^2 F)_i + \lambda I$. The difference of all baseline methods lies only in the process of finding approximate Newton directions in each iteration. In iteration $t$, GIANT (Wang et al., 2017), Determinantal Averaging (Dereziński & Mahoney, 2019) ("Determinant" below), Optimal Shrinkage (Zhang & Pilanci, 2023) ("Shrinkage" below), and DiSCO (Zhang & Lin, 2015) methods all take the next Newton step as $x^{(t+1)} = x^{(t)} - \eta^{(t)} \hat{H}^{(t)^{-1}} g^{(t)}$ where $\eta^{(t)}$ is the step size found by line search in current iteration, $g^{(t)}$ is the global gradient and $\hat{H}^{(t)}$ is an estimate of the Hessian at $x^{(t)}$. Specifically,

- GIANT takes estimated Hessian inverse as average over all local Hessian inverses, i.e., $\hat{H}^{(t)^{-1}} = \frac{1}{q} \sum_i \left((\nabla^2 F)_i^t + \lambda I\right)^{-1}$.

- Determinantal Averaging exploits the expectation identity $\mathbb{E}(\det(A))(\mathbb{E}(A))^{-1} = E(\det(A)A^{-1})$ which is valid under mild conditions for invertible $A$ and approximates the Hessian inverse by $\hat{H}^{(t)^{-1}} = \frac{\sum_i \det((\nabla^2 F)_i^t + \lambda I)((\nabla^2 F)_i^t + \lambda I)^{-1}}{\sum_i \det((\nabla^2 F)_i^t + \lambda I)}$.

- Optimal Shrinkage method approximates Hessian inverse with $\hat{H}^{(t)^{-1}} = \frac{1}{q} \sum_i \left(\frac{1}{1 - \frac{d_\lambda^t}{n/q}}(\nabla^2 F)_i^t + \lambda I\right)^{-1}$ where $d_\lambda^t$ is the effective dimension of $(\nabla^2 F)^t$ and it requires $n/q > d_\lambda^t$. In the experiments, we follow the approach in (Zhang & Pilanci, 2023) and take $d_\lambda^t$ used by worker $i$ as effective dimension of local $(\nabla^2 F)_i^t$.

- DiSCO approximates Hessian inverse by using the first worker's statistics $\hat{H}^{(t)^{-1}} = ((\nabla^2 F)_1^t + \lambda I)^{-1}$.

- DANE considers a slightly different update rule which involves solving

$$x_i^{(t+1)} = \operatorname{argmin}_x F_i(x) + \frac{\lambda}{2}\|x\|_2^2 - ((\nabla F_i(x^{(t)}) + \lambda x^{(t)}) - \alpha g^{(t)})^T x + \frac{\beta}{2}\|x - x^{(t)}\|_2^2,$$

---

[9]Here, we assume $\frac{n}{q}$ is an integer. For real-world datasets, any remainder when dividing $n$ by $q$ is disregarded for simplicity.

then it takes the next iterate as $x^{(t+1)} = \frac{1}{q} \sum_i x_i^{(t+1)}$. We take $\alpha = 1$ and $\beta = \frac{1}{2}$ in our experiments.

We compare these methods across two scenarios, namely ridge regression and logistic regression, detailed in the following subsections.

### A.5.2 RIDGE REGRESSION

Due to the varying sizes of samples in each dataset, we set number of workers to $q = 10$ for "bodyfat", "splice", and "pyrim" datasets; $q = 20$ for "segment", "eunite2001" and "iris" datasets; $q = 30$ for "mg" dataset. For each method, we repeat the optimization process over 10 random seeds, with the data split (or sketch) redrawn for each iteration, and plot the mean and one standard deviation of the logarithmic optimality gap. Figure 5 shows the convergence progress for different datasets. Since for quadratic loss, DANE and GIANT coincide, we thus write "DANE/GIANT" to represent both methods. It can be observed that our adaptive de-biased Hessian inverse estimation accelerates convergence significantly and converges the fastest in almost all experiments. The most significant improvement over baseline methods happens on "eunite2001", "bodyfat", and "pyrim" datasets, where all baseline methods take at least $\sim 20$ iterations to reach certain accuracy and our method takes $\leq 5$ iterations. On "segment" and "mg" datasets, though most baseline methods are already good and converge within 10 steps, our de-biased Hessian inverse estimator still provides improvement in this case. Moreover, our method exhibits the smallest variance, which highlights its robustness and stability.

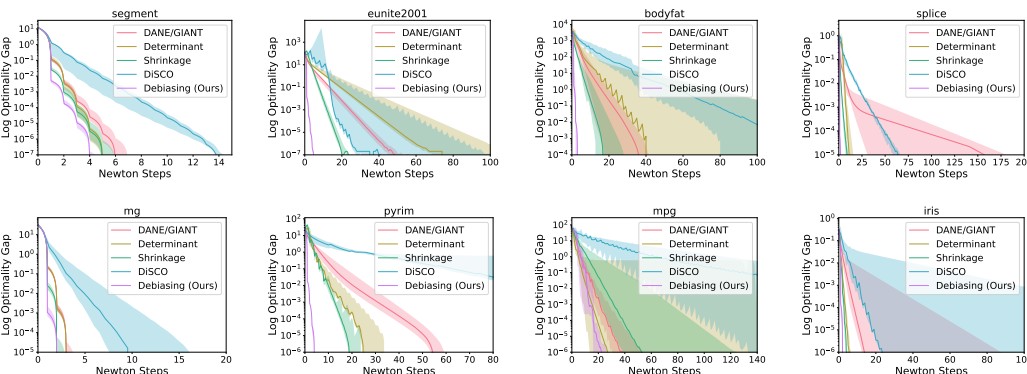

Figure 5: Comparison of Distributed Newton Methods for Ridge Regression: Our method demonstrates the fastest convergence and the lowest variance across nearly all experiments, highlighting its effectiveness and robustness.

### A.5.3 LOGISTIC REGRESSION

Here we set $q = 5$ for "liver-disorders" and "ionosphere" datasets; $q = 10$ for "heart" dataset; $q = 20$ for "diabetes" and "sonar" datasets; $q = 50$ for "german_numer" dataset; $q = 250$ for "fourclass" dataset; $q = 500$ for "svmguide3" dataset. For each method, we repeat the optimization process over 10 random seeds, with the data split (or sketch) redrawn for each iteration, and plot the mean and one standard deviation of the logarithmic optimality gap. Figure 6 shows the convergence progress for various methods on different datasets. We observe that the convergence behavior for logistic regression across different methods is more challenging to achieve compared to ridge regression. This difference may be attributed to the non-quadratic nature of the logistic regression objective function. From the results, one can observe that our de-biasing method always converges faster than all baseline methods and the improvement is most significant on "german_numer", "ionosphere" and "sonar" datasets where our method halves the iteration number compared to the second best method. Moreover, DANE diverges on datasets such as "german_numer" and "fourclass", Determinantal averaging encountered NaN values during optimization and we thus skip it for "ionosphere" and "sonar" datasets. This may be due to the numerical issues in the determinant calculations. Although most of the methods suffer from high variability, our method still behaves stably with little variation. This again proves the robustness of our Hessian inverse de-biaser.

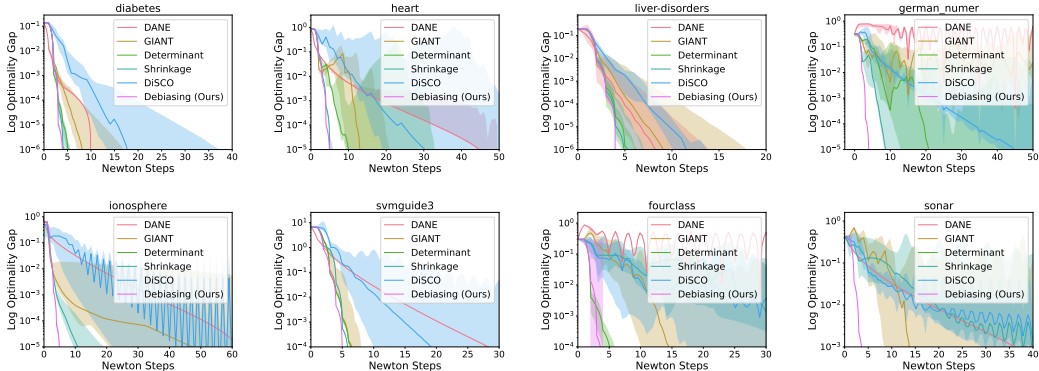

Figure 6: Comparison with distributed Newton methods for logistic regression. Our method converges the fastest and exhibits the smallest variance for almost all experiments, which shows its effectiveness and robustness.

# B  NON-ASYMPTOTIC RANDOM MATRIX THEORY RESULTS

The proofs of Theorems 3.4-3.6, rely on non-asymptotic results from random matrix theory that that we state in this section and prove later on. The results below concern the behavior of the random matrices $SHS^\top, H^{1/2}S^\top SH^{1/2}$ where $H \succeq 0$ and $S \in \mathbb{R}^{m \times d}$ is an i.i.d. Gaussian matrix $S_{ij} \sim \mathcal{N}(0, 1/m)$.

The following result concerns the smallest eigenvalue of the matrix $SHS^\top \in \mathbb{R}^{m \times m}$.

**Theorem B.1.** *Suppose $S$ has i.i.d. Gaussian entries. There are constants $c_1, c_2, c, C > 0$ such that the following holds. Set any $\eta > 0$. If $m \leq c_1 \mathsf{d}_H(\eta)$ then w.p. $1 - Ce^{-cm}$,*

$$\lambda_{\min}(SHS^\top) \geq c_2 \eta \frac{1}{m} \mathsf{d}_H(\eta).$$

The following two results concern the closedness of the empirical (companion) Stieltjes transform

$$\hat{\mathsf{s}}(z) := \frac{1}{m} \operatorname{tr}\left(SHS^\top - zI\right)^{-1}, \tag{26}$$

to the deterministic function $\mathsf{s}(z)$, the solution of the Marchenko-Pastur equation (10).

The following concentration inequality is well known, cf. Bai & Silverstein (2010).

**Lemma B.2.** *Suppose that $S = \frac{1}{\sqrt{m}}[r_1| \cdots |r_m]^\top$ where the rows $r_1, \dots, r_m \in \mathbb{R}^d$ are independent random variables. For any $z < 0$ and $t \geq 0$,*

$$\Pr(|\hat{\mathsf{s}}(z) - \mathbb{E}\hat{\mathsf{s}}(z)| \geq t) \leq 2e^{-cmz^2t^2}.$$

We provide a self-contained proof for completeness, see Section F.

Next, we show that $\mathbb{E}\hat{\mathsf{s}}(z)$ approximately satisfies (10), with explicit non-asymptotic error bounds. Note that (10) can be be equivalently rewritten (assuming $z \neq 0$) as

$$\mathsf{s}(z) = -\frac{1}{z}\left(1 - \frac{1}{m}\mathsf{d}_H(1/\mathsf{s}(z))\right). \tag{27}$$

**Theorem B.3.** *Suppose that $S$ has i.i.d. Gaussian entries. For any $z < 0$, $\mathbb{E}\hat{\mathsf{s}}(z)$ satisfies:*

$$\mathbb{E}\hat{\mathsf{s}}(z) = -\frac{1}{z}\left(1 - \frac{1}{m}\mathsf{d}_H(1/\mathbb{E}\hat{\mathsf{s}}(z))\right) + e_m(z; H), \tag{28}$$

*where the error term $e_m(z; H)$ satisfies*

$$e_m(z; H) \lesssim \frac{\beta}{\alpha} \frac{1}{|z|\sqrt{m}}, \tag{29}$$

*with $\alpha = \alpha_m(z; H), \beta = \beta_m(z; H)$ defined as*

$$\alpha := |z| \mathbb{E}\hat{\mathsf{s}}(z) \, , \tag{30}$$

$$\beta := \frac{1}{m} \mathsf{d}_H(1/\mathbb{E}\hat{\mathsf{s}}(z)) \, . \tag{31}$$

The proof of Theorem B.3, given in Section G, follows along a well-known computation technique in random matrix theory, and explicitly leverages the orthogonal invariance of the Gaussian distribution. Note that the error terms $\alpha, \beta$ depend on $\mathbb{E}\hat{\mathsf{s}}(z)$ itself. This feature, which somewhat complicates the proofs of Theorems 3.4-3.5, is generally not a mere artifact of the calculation, see e.g. the bounds of (Knowles & Yin, 2017) under asymptotic conditions.

## C  PROOF OF THEOREM 3.4

Denote for brevity, per the description of Algorithm 1,

$$z_0 := -\frac{5}{12}\lambda. \tag{32}$$

The proof of Theorem 3.4 consists of two parts. First, we show that if $m < 1.5\mathsf{d}_H(\lambda)$, then with high probability (w.h.p.) $\hat{\mathsf{s}}_m(z_0) < 1/\lambda$, hence such $m$ would not be returned. Then, we show that when $m \geq 2\mathsf{d}_H(\lambda)$, then w.h.p. $\hat{\mathsf{s}}_m(z_0) \geq 1/\lambda$, which would guarantee that the algorithm must return such $m$ (if it has not halted before). Consequently, the output of the algorithm must satisfy $m \leq 4\mathsf{d}_H(\lambda)$.

The next lemma shows that Algorithm 1 does not return $m$ which is *much* smaller than $\mathsf{d}_H(\lambda)$.

**Lemma C.1.** *Suppose that $S$ has i.i.d. Gaussian entries. There is a (small) constant $c_0$ such that if $m < c_0\mathsf{d}_H(\lambda)$ then*

$$\Pr(\hat{\mathsf{s}}(z_0) > 1/\lambda) \leq Ce^{-cm}.$$

*Proof.* The lemma is a consequence of Theorem B.1, which bounds the smallest eigenvalue of $SHS^\top$. Set $c_0 = \min\{c_1, 12c_2/7\}$ for $c_1, c_2$ from Theorem B.1. Assume that $m < c_0\mathsf{d}_H(\lambda)$. Since in particular $m < c_1\mathsf{d}_H(\lambda)$, the following event holds w.p. $1 - Ce^{-cm}$:

$$\hat{\mathsf{s}}(z_0) \leq \frac{1}{\lambda_{\min}(SHS^\top) - z_0} \leq \frac{1}{c_2\frac{1}{m}\mathsf{d}_H(\lambda) \cdot \lambda + \frac{5}{12}\lambda}.$$

Moreover, since $c_2/c_0 \geq 7/12$,

$$c_2\frac{1}{m}\mathsf{d}_H(\lambda) > c_2\frac{1}{c_0} \geq \frac{7}{12}.$$

Hence, under this event, $\hat{\mathsf{s}}(z_0) < 1/\lambda$. $\qquad\square$

Next, we treat the regime of moderate $m$, namely when $c_0\mathsf{d}_H(\lambda) \leq m < 1.5\mathsf{d}_H(\lambda)$. The main tool here is the non-asymptotic estimate of Theorem B.3. The reason we have to assume $m \gtrsim \mathsf{d}_H(\lambda)$ is that only then we can guarantee the error term is controlled.

The following simple lemma is useful.

**Lemma C.2.** *For any $\mu > 0$ and $\gamma \geq 1$,*

$$\mathsf{d}_H(\gamma\mu) \leq \mathsf{d}_H(\mu) \leq \gamma\mathsf{d}_H(\gamma\mu). \tag{33}$$

*Proof.* The inequality $\mathsf{d}_H(\gamma\mu) \leq \mathsf{d}_H(\mu)$ is trivial. As for the other one,

$$\mathsf{d}_H(\gamma\mu) = \mathrm{tr}(H(H + \gamma\mu I)^{-1}) = \frac{1}{\gamma}\,\mathrm{tr}(H(\frac{1}{\gamma}H + \mu I)^{-1}) \overset{(\star)}{\geq} \frac{1}{\gamma}\,\mathrm{tr}(H(H + \mu I)^{-1}) = \frac{1}{\gamma}\mathsf{d}_H(\mu),$$

where $(\star)$ follows because $H \succeq 0$, hence $\frac{1}{\gamma}H + \mu I \preceq H + \mu I$ and therefore $H(\frac{1}{\gamma}H + \mu I)^{-1} \succeq H(H + \mu I)^{-1}$. $\qquad\square$

**Lemma C.3.** *Suppose that $S$ has i.i.d. Gaussian entries. Moreover, suppose that $c_0\mathsf{d}_H(\lambda) \leq m < 1.5\mathsf{d}_H(\lambda)$. There is a numerical constant $M$ such that if moreover $m \geq M$ then*

$$\mathbb{E}\hat{\mathsf{s}}(z_0) \leq \frac{14}{15} \cdot \frac{1}{\lambda}.$$

*In particular, $\Pr(\hat{\mathsf{s}}(z_0) > 1/\lambda) \leq Ce^{-cm}$.*

*Proof.* Note that the second part follows directly from the first part and Lemma B.2; we therefore focus on bounding $\mathbb{E}\hat{\mathsf{s}}(z_0)$. Suppose it were the case that $\mathbb{E}\hat{\mathsf{s}}_m(z_0) \geq \eta\frac{1}{\lambda}$ where $\eta \leq 1$ is some constant. Apply Theorem B.3 with $z = z_0$. Let us bound the parameters $\alpha, \beta$ used for error control. First,

$$\alpha := |z_0|\mathbb{E}\hat{\mathsf{s}}(z_0) \geq \frac{5}{12}\lambda \cdot \eta\frac{1}{\lambda} \geq \frac{5}{12}\eta. \tag{34}$$

Next, since $\mathbb{E}\hat{s}(z_0) \leq \frac{1}{|z_0|} = \frac{12}{5}\frac{1}{\lambda}$ we have $1/\mathbb{E}\hat{s}(z_0) \geq \frac{5}{12}\lambda$, so

$$\beta := \frac{1}{m}\mathsf{d}_H(1/\mathbb{E}\hat{s}(z_0)) \leq \frac{1}{m}\mathsf{d}_H(\frac{5}{12}\lambda) \overset{(\star)}{\leq} \frac{12}{5}\frac{1}{m}\mathsf{d}_H(\lambda) \overset{(\star\star)}{\leq} \frac{12}{5}\frac{1}{c_0}, \tag{35}$$

where $(\star)$ follows from Lemma C.2 and $(\star\star)$ from the assumption $m \geq c_0\mathsf{d}_H(\lambda)$. In particular, the error $e_m(z_0, H)$ in (29) can be bound as $e_m(z_0, H) = O_\eta(\frac{1}{\lambda\sqrt{m}})$. Furthermore, as $\mathbb{E}\hat{s}(z_0) \geq \eta\frac{1}{\lambda}$ by assumption,

$$\beta \geq \frac{1}{m}\mathsf{d}_H(\frac{1}{\eta}\lambda) \geq \eta\frac{1}{m}\mathsf{d}_H(\lambda) > \frac{2}{3}\eta, \tag{36}$$

where we used Lemma C.2 and the assumption $m < 1.5\mathsf{d}_H(\lambda)$. With these estimates in mind, consider (28),

$$\mathbb{E}\mathsf{s}(z_0) = -\frac{1}{z_0}(1 - \beta - z_0\varepsilon_m(z_0, H)) = \frac{1}{\lambda}\cdot\frac{12}{5}(1 - \beta - z_0 e_m(z_0, H)) \leq \frac{1}{\lambda}\cdot\frac{12}{5}(1 - \frac{2}{3}\eta + O_\eta(\frac{1}{\sqrt{m}})).$$

As $\mathbb{E}\hat{s}_m(z_0) \geq \eta\frac{1}{\lambda}$, we deduce from the above

$$\eta \leq \frac{12}{5}(1 - \frac{2}{3}\eta + O_\eta(\frac{1}{\sqrt{m}})),$$

so that rearranging,

$$\eta \leq \frac{12}{13}(1 + O_\eta(\frac{1}{\sqrt{m}})).$$

By requiring that $m \geq C_\eta$ for large enough $C_\eta$, we can ensure that the error term is as small a constant as we like, so that necessarily (for example) $\eta \leq \frac{14}{15}$.

$\square$

Next, we show that if the algorithm has reached $m \geq 2\mathsf{d}_H(\lambda)$, then it will halt in that iteration. We start with two helper lemmas.

**Lemma C.4.** *Suppose that $\mathbb{E}[S^\top S] = I$. Then for all $z < 0$,*

$$\mathbb{E}\hat{s}(z) \geq -\frac{1}{z}\left(1 - \frac{1}{m}\mathsf{d}_H(-z)\right). \tag{37}$$

*(Note that since $\mathbb{E}\hat{s}(z) \leq -1/z$, we have $\mathsf{d}_H(-z) \geq \mathsf{d}_H(1/\mathbb{E}\hat{s}(z))$, so that (37) is perfectly consistent with (28).)*

*Proof.* Recall that the matrices $SHS^\top \in \mathbb{R}^{m\times m}$ and $H^{1/2}S^\top SH^{1/2}$ have the same non-zero eigenvalues, and therefore

$$\mathbb{E}\hat{s}(z) := \frac{1}{m}\mathbb{E}\operatorname{tr}(SHS^\top - zI)^{-1} = \frac{1}{m}\mathbb{E}\operatorname{tr}(H^{1/2}S^\top SH^{1/2} - zI)^{-1} + \frac{1}{m}(d - m)\frac{1}{z}.$$

Note that since $z < 0$, the matrix function $A \mapsto \operatorname{tr}(A - zI)^{-1}$ is convex on the cone of PSD matrices $A \succeq 0$. Thus, by Jensen's inequality,

$$\frac{1}{m}\mathbb{E}\operatorname{tr}(H^{1/2}S^\top SH^{1/2} - zI)^{-1} \geq \frac{1}{m}\operatorname{tr}(\mathbb{E}[H^{1/2}S^\top SH^{1/2}] - zI)^{-1} = \frac{1}{m}\operatorname{tr}(H - zI)^{-1},$$

so that

$$\mathbb{E}\hat{s}(z) \geq \frac{1}{m}\operatorname{tr}(H - zI)^{-1} + \frac{1}{m}(d - m)\frac{1}{z} = \frac{1}{m}\operatorname{tr}[(H - zI)^{-1} + \frac{1}{z}I] - \frac{1}{z} = -\frac{1}{z}\left(1 - \frac{1}{m}\mathsf{d}_H(-z)\right).$$

$\square$

The next lemma formally establishes that the inverse function of $\mathsf{s}(z)$ is increasing on an appropriate interval. As in (11), denote $\Psi : (0, \infty) \to \mathbb{R}$,

$$\Psi(s) = -\frac{1}{s}(1 - \frac{1}{m}\mathsf{d}_H(1/s)), \tag{38}$$

so that for all $z < 0$, $\Psi(\mathsf{s}(z)) = z$. Note however that in general, $\operatorname{Range}(\mathsf{s})$ may only be a proper subset of $(0, \infty)$; nonetheless, (38) is well-defined.

**Lemma C.5.** *Let $\mu > 0$, and suppose that $m \geq \mathsf{d}_H(\mu)$. Then $\Psi$ is increasing on $(0, 1/\mu)$.*

*Proof.* Write

$$\Psi(s) = -\frac{1}{s} + \frac{1}{m}\operatorname{tr}(H(I + sH)^{-1}),$$

so that its derivative is

$$
\begin{aligned}
\Psi'(s) &= \frac{1}{s^2} - \frac{1}{m}\operatorname{tr}(H^2(I + sH)^{-2}) \\
&= \frac{1}{s^2}\left(1 - \frac{1}{m}\operatorname{tr}(H^2(H + 1/s)^{-2})\right) \\
&\geq \frac{1}{s^2}\left(1 - \frac{1}{m}\operatorname{tr}(H(H + 1/s)^{-1})\right) = \frac{1}{s^2}\left(1 - \frac{1}{m}\mathsf{d}_H(1/s)\right),
\end{aligned}
\tag{39}
$$

where the inequality follows since $0 \preceq H(H + 1/s)^{-1} \preceq I$. When $s < 1/\mu$, we have $1/s > \mu$ so that $\mathsf{d}_H(1/s) < \mathsf{d}_H(\mu)$, hence $\Psi'(s) > 0$. $\square$

We now prove:

**Lemma C.6.** *Suppose that $S$ has i.i.d. Gaussian entries, and moreover that $m \geq 2\mathsf{d}_H(\lambda)$. There is a numerical $M$ such that whenever $m \geq M$,*

$$\mathbb{E}\hat{\mathsf{s}}(z_0) \geq \frac{24}{23} \cdot \frac{1}{\lambda}.$$

*In particular, $\Pr(\hat{\mathsf{s}}(z_0) \leq 1/\lambda) \leq Ce^{-cm}$.*

*Proof.* Suppose that $\mathbb{E}\hat{\mathsf{s}}(z_0) \leq \frac{1}{(1-\eta)}\frac{1}{\lambda}$ for some $\eta \in (0, 1)$; we shall show that $\eta$ has to be somewhat large so to not generate a contradiction. First, by Lemma C.4 (with $z = -\lambda$),

$$\mathbb{E}\hat{\mathsf{s}}(z_0) \geq \mathbb{E}\hat{\mathsf{s}}(-\lambda) = \frac{1}{\lambda}(1 - \frac{1}{m}\mathsf{d}_H(\lambda)) = \frac{1}{2} \cdot \frac{1}{\lambda},$$

where the last inequality follows since $m \geq 2\mathsf{d}_H(\lambda)$. Of course, this lower bound does not suffice for our purposes: we want a bound which is $> \frac{1}{\lambda}$ (strictly). It does allow us, however, to control the error term in Theorem B.3 at $z = z_0$ (specifically, to lower bound $\alpha$). We can bound the parameters $\alpha, \beta$ in Theorem B.3 as

$$\alpha := |z_0|\mathbb{E}\hat{\mathsf{s}}(z_0) \geq \frac{5}{12}\lambda \cdot \frac{1}{2}\frac{1}{\lambda} \geq \frac{5}{24},$$

$$\beta := \frac{1}{m}\mathsf{d}_H(1/\mathbb{E}\hat{\mathsf{s}}(z_0)) \leq \frac{1}{m}\mathsf{d}_H(-z_0) = \frac{1}{m}\mathsf{d}_H(\frac{5}{12}\lambda) \leq \frac{12}{5}\frac{1}{m}\mathsf{d}_H(\lambda) \leq \frac{6}{5},$$

where, in upper bounding $\beta$, we used $\mathbb{E}\hat{\mathsf{s}}(z_0) \leq \frac{1}{-z_0}$ and Lemma C.2. With these, Theorem B.3 implies

$$\mathbb{E}\hat{\mathsf{s}}(z_0) = -\frac{1}{z_0}(1 - \frac{1}{m}\mathsf{d}_H(1/\mathbb{E}\hat{\mathsf{s}}_m(z_0))) + O(\frac{1}{\lambda\sqrt{m}}).$$

Multiplying by $z_0$ and dividing by $\mathbb{E}\hat{\mathsf{s}}(z_0)$,

$$
\begin{aligned}
z_0 &= -\frac{1}{\mathbb{E}\hat{\mathsf{s}}(z)}\left(1 - \frac{1}{m}\mathsf{d}_H(1/\mathbb{E}\hat{\mathsf{s}}(z))\right) + O(\frac{\lambda}{\sqrt{m}}) \\
&= \Psi(\mathbb{E}\hat{\mathsf{s}}(z)) + O(\frac{\lambda}{\sqrt{m}}),
\end{aligned}
$$

where in modifying the error term we used $\mathbb{E}\hat{\mathsf{s}}(z_0) \gtrsim \frac{1}{\lambda}$.

Note that $m \geq 2\mathsf{d}_H(\lambda)$ implies $m \geq \mathsf{d}_H(\frac{1}{2}\lambda)$ by Lemma C.2. By Lemma C.5, this implies that $\Psi$ is increasing on $(0, 2\frac{1}{\lambda})$. Thus, if we further assume that $\eta < 1/2$, then $\Psi$ is increasing on $(0, \frac{1}{1-\eta}\frac{1}{\lambda})$,

and so $\Psi(\mathbb{E}\hat{s}(z)) \leq \Psi(\frac{1}{1-\eta}\frac{1}{\lambda})$. Using this with the previous display, and writing $z_0 = -\frac{5}{12}\lambda$,

$$-\frac{5}{12}\lambda \leq \Psi(\frac{1}{1-\eta}\frac{1}{\lambda}) + O(\frac{\lambda}{\sqrt{m}})$$

$$= -(1-\eta)\lambda\left(1 - \frac{1}{m}\mathsf{d}_H((1-\eta)\lambda)\right) + O(\frac{\lambda}{\sqrt{m}})$$

$$\overset{(\star)}{\leq} -(1-\eta)\lambda\left(1 - \frac{1}{1-\eta}\frac{1}{m}\mathsf{d}_H(\lambda)\right) + O(\frac{\lambda}{\sqrt{m}})$$

$$\overset{(\star\star)}{\leq} -(1-\eta)\lambda\left(1 - \frac{1}{1-\eta}\frac{1}{2}\right) + O(\frac{\lambda}{\sqrt{m}})$$

$$= -\lambda(\frac{1}{2} - \eta) + O(\frac{\lambda}{\sqrt{m}}),$$

where $(\star)$ uses $\mathsf{d}_H((1-\eta)\lambda) \leq \frac{1}{1-\eta}\mathsf{d}_H(\lambda)$ (Lemma C.2) and $(\star\star)$ uses the assumption $m \geq 2\mathsf{d}_H(\lambda)$. Dividing by $(-\lambda)$,

$$\frac{5}{12} \geq \frac{1}{2} - \eta + O(\frac{1}{\sqrt{m}}).$$

For large enough $m \geq M$, the $O(\cdot)$ term is $\leq \frac{1}{24}$, therefore $\eta \geq \frac{1}{24}$. That is, we deduce that necessarily $\mathbb{E}\hat{s}(z_0) \geq \frac{1}{1-\frac{1}{24}}\frac{1}{\lambda} = \frac{24}{23}\frac{1}{\lambda}$. □

We are ready to conclude the proof of Theorem 3.4.

*Proof.* (Of Theorem 3.4.) Let $m_j = 2^j m_0$, $j = 0, 1, \ldots$, be the value of $m$ at iteration $j$ of the algorithm. Set

$$J^- = \lceil \log_2(1.5\mathsf{d}_H(\lambda)/m_0)\rceil, \qquad J^+ = \lceil \log_2(2\mathsf{d}_H(\lambda)/m_0)\rceil,$$

chosen so that: 1) $J^-$ is the smallest $j$ such that $m_j \geq 1.5\mathsf{d}_H(\lambda)$; 2) $2\mathsf{d}_H(\lambda) \leq m_{J^+} < 4\mathsf{d}_H(\lambda)$. Thus, to establish (19) it suffices to show that the algorithm halts at an iteration $j$ satisfying $J^- \leq j \leq J^+$. Accordingly, the error probability can be bounded as

$$\Pr((19) \text{ does not hold}) \leq \sum_{j=0}^{J^--1} \Pr(\hat{s}_{m_j}(z_0) > 1/\lambda) + \Pr(\hat{s}_{m_{J^+}}(z_0) \leq 1/\lambda).$$

Assume that $m_0 \geq M$ is large enough. By Lemmas C.1 and C.3, for all $j \leq J^- - 1$, $\Pr(\hat{s}_{m_j}(z_0) > 1/\lambda) \leq Ce^{-cm_j}$. By Lemma C.6, $\Pr(\hat{s}_{m_{J^+}}(z_0) \leq 1/\lambda) \leq Ce^{-cm_{J^+}}$. Thus,

$$\Pr((19) \text{ does not hold}) \leq \sum_{j=0}^{J^--1} Ce^{-c2^j m_0} + Ce^{-c2^{J^+} m_0} \leq \sum_{j=0}^{\infty} Ce^{-c2^j m_0} \lesssim e^{-cm_0}.$$

We deduce that if $m_0 \gtrsim \log(1/\delta)$, then the error probability is $\leq \delta$. □

# D  PROOF OF THEOREM 3.5

Recall the (deterministic) Marchenko-Pastur equation (10); rearranging yields

$$z = -\frac{1}{\mathsf{s}(z)}\left(1 - \frac{1}{m}\mathsf{d}_H(1/\mathsf{s}(z))\right),\tag{40}$$

so that setting $\mathsf{s}(-\tilde{\lambda}) = 1/\lambda$ yields $\tilde{\lambda} = \lambda(1 - \frac{1}{m}\mathsf{d}_H(\lambda))$. Further recall that we do not have access to $\mathsf{s}(z)$, but instead to its random counterpart $\hat{\mathsf{s}}(z)$. We know that its expectation $\mathbb{E}\hat{\mathsf{s}}(z)$ approximately satisfies the Marcheko-Pastur equation with an error term (Theorem B.3), and that we have concentration around the expectation (Lemma B.2). The proof of Theorem 3.5 down to controlling and propagating these error terms.

**Lemma D.1.** *Suppose that $m \geq 1.5\mathsf{d}_H(\lambda)$ and $m \geq M$ for large enough $M$.*

*A unique root $z^\star < 0$ such that $\mathbb{E}\hat{\mathsf{s}}(z^\star) = \frac{1}{\lambda}$ exists, and $z^\star < -c\lambda$ for small enough $c > 0$.*

*Consider a sufficiently small $O(\lambda)$-neighborhood of $z^\star$, $I^\star = [z^\star - \eta\lambda, z^\star + \eta\lambda]$ for small enough $\eta > 0$. Then,*

1. *$\mathbb{E}\hat{\mathsf{s}}(z) \gtrsim \frac{1}{\lambda}$ for all $z \in I^\star$.*

2. *$|\mathbb{E}\hat{\mathsf{s}}(z) - \mathbb{E}\hat{\mathsf{s}}(z')| \lesssim |z - z'|\frac{1}{\lambda^2}$ for $z, z' \in I^\star$.*

*Proof.* The existence of $z^\star$, as well as the bound $\mathbb{E}\hat{\mathsf{s}}_m(z) \geq \eta\frac{1}{\lambda}$, can be shown by an argument similar to Lemma C.6; we omit the technical details. Specifically, one can show that for small enough $c$, $\mathbb{E}\hat{\mathsf{s}}(-c\lambda) \geq (1+c)\frac{1}{\lambda}$. Since $\mathbb{E}\mathsf{s}(\cdot)$ is continuous increasing, and $\mathbb{E}\hat{\mathsf{s}}(-\infty) = 0$, a root $\mathbb{E}\hat{\mathsf{s}}(z^\star) = \frac{1}{\lambda}$ exists and $z^\star < -c\lambda$.

Next, we prove Item 2 above. We have

$$|\mathbb{E}\hat{\mathsf{s}}_m(z) - \mathbb{E}\hat{\mathsf{s}}_m(z')| = \mathbb{E}\frac{1}{m}\operatorname{tr}[(S_m H S_m^\top - zI)^{-1}(S_m H S_m^\top - z'I)^{-1}]|z - z'| \leq \frac{|z - z'|}{|zz'|} \lesssim \frac{1}{\lambda^2}|z - z'|,$$

since $|z|, |z'| \gtrsim \lambda$, as $z, z' \in I_\eta$.

Item 1 follows immediately from Item 2. $\qquad\square$

We need to establish a result of the following kind: if $z < 0$ is such that $\mathbb{E}\hat{\mathsf{s}}(z) \approx \frac{1}{\lambda}$, then necessarily $z \approx z^\star$. This necessitates a lower bound (rather than an upper bound, such as given in Lemma D.1, Item 2) on the derivative of $\mathbb{E}\hat{\mathsf{s}}(z)$ around $z^\star$.

**Lemma D.2.** *Suppose that $m \geq 1.5\mathsf{d}_H(\lambda)$ and $m \geq M$ for large enough $M$.*

*For a sufficiently small $O(\lambda)$-neighborhood of $z^\star$, $z^\star \in I^\star$,*

$$|\mathbb{E}\hat{\mathsf{s}}(z) - \mathbb{E}\hat{\mathsf{s}}(z')| \gtrsim \frac{1}{\lambda^2}|z - z'| - O(\frac{1}{\lambda\sqrt{m}}).\tag{41}$$

*Proof.* By Theorem B.3, having established that $\mathbb{E}\hat{\mathsf{s}}(z) \gtrsim \frac{1}{\lambda}$ for all $z \in I^\star$, uniformly for all $z \in I^\star$,

$$\mathbb{E}\hat{\mathsf{s}}(z) = -\frac{1}{z}\left(1 - \frac{1}{m}\mathsf{d}_H(1/\mathbb{E}\hat{\mathsf{s}}(z))\right) + O(\frac{1}{\lambda\sqrt{m}}).$$

uniformly for all $z \in I^\star$. Dividing by $\mathbb{E}\hat{\mathsf{s}}(z)$ and multiplying by $z$,

$$z = \Psi(\mathbb{E}\hat{\mathsf{s}}(z)) + O(\frac{\lambda}{\sqrt{m}}),\tag{42}$$

where $\Psi(s) = -\frac{1}{s}(1 - \frac{1}{m}\mathsf{d}_H(1/s))$ is the inverse function of $\mathsf{s}(z)$ (see e.g. (38)). Note that for $z, z' \in I^\star$, $|\mathbb{E}\hat{\mathsf{s}}(z) - \mathbb{E}\hat{\mathsf{s}}(z')| \lesssim |z - z'|\frac{1}{\lambda^2} = O(\frac{1}{\lambda})$. Thus, to conclude the proof of the lemma, it suffices to give an upper bound on the derivative (or Lipschitz constant) of $\Psi$ in a small $O(1/\lambda)$ neighborhood of $\mathbb{E}\hat{\mathsf{s}}(z^\star) = \frac{1}{\lambda}$.

Indeed, if $s = \frac{1}{\lambda}(1 + \varepsilon)$ for small $\varepsilon = O(1)$, then by Lemma C.2, $\frac{1}{1+|\varepsilon|}\mathsf{d}_H(\lambda) \leq \mathsf{d}_H(1/s) \leq \frac{1}{1-|\varepsilon|}\mathsf{d}_H(\lambda)$. Since $\frac{1}{m}\mathsf{d}_H(\lambda) \leq (1.5)^{-1} = O(1)$, $|\frac{1}{m}\mathsf{d}_H(1/s) - \frac{1}{m}\mathsf{d}_H(1/\lambda)| = O(\varepsilon)$. Thus, for $s_1, s_2$ in a sufficiently small $O(\frac{1}{\lambda})$-neighborhood of $\frac{1}{\lambda}$, $|\frac{1}{m}\mathsf{d}_H(1/s_1) - \frac{1}{m}\mathsf{d}_H(1/s_2)| = O(\lambda|s_1 - s_2|)$, and so $|\Psi(s_1) - \Psi(s_2)| = O(\lambda^2|s_1 - s_2|)$. Using (42), we deduce that for $z_1, z_2$ in a small enough $O(\lambda)$ neighborhood of $z^\star$,

$$|z_1 - z_2| = \left|\Psi(\mathbb{E}\hat{\mathsf{s}}(z_1)) - \Psi(\mathbb{E}\hat{\mathsf{s}}(z_2)) + O(\frac{\lambda}{\sqrt{m}})\right|$$

$$\lesssim \lambda^2|\mathbb{E}\hat{\mathsf{s}}(z_1) - \mathbb{E}\hat{\mathsf{s}}(z_2)| + O(\frac{\lambda}{\sqrt{m}}).$$

The desired estimate (41) follows dividing by $\lambda^2$. $\qquad\square$

We are ready to conclude the proof of Theorem 3.5. Let $\varepsilon > 0$, $\delta \in (0,1)$ be, respectively, the precision on confidence parameters. Suppose that $\varepsilon = O(1)$ is small, such that $[z^\star - 2\lambda\varepsilon, z^\star + 2\lambda\varepsilon] \subseteq I^\star$, therefore setting

$$z^\star_\pm = z^\star \pm \lambda\varepsilon \tag{43}$$

we have $z^\star_\pm \in I^\star$. By Lemma D.2, Eq. (41), provided that $m \gtrsim 1/\varepsilon^2$ is large enough,

$$\mathbb{E}\hat{\mathsf{s}}(z^\star_-) \leq \frac{1}{\lambda} - C_1\varepsilon\frac{1}{\lambda} + O(\frac{1}{\lambda\sqrt{m}}) \leq \frac{1}{\lambda} - C_2\varepsilon\frac{1}{\lambda},$$

$$\mathbb{E}\hat{\mathsf{s}}(z^\star_+) \geq \frac{1}{\lambda} + C_1\varepsilon\frac{1}{\lambda} - O(\frac{1}{\lambda\sqrt{m}}) \geq \frac{1}{\lambda} + C_2\varepsilon\frac{1}{\lambda}.$$

By Lemma B.2, if $m \gtrsim \frac{1}{\varepsilon^2}\log(1/\delta)$ is large enough, then w.p. $1 - \delta$, $|\hat{\mathsf{s}}(z^\star_-) - \mathbb{E}\hat{\mathsf{s}}(z^\star_-)|, |\hat{\mathsf{s}}(z^\star_+) - \mathbb{E}\hat{\mathsf{s}}(z^\star_+)| \leq C_2\frac{1}{\lambda}\varepsilon/2$. On this event,

$$\hat{\mathsf{s}}(z^\star_-) \leq \frac{1}{\lambda} - \frac{1}{2}C_2\varepsilon\frac{1}{\lambda} < \frac{1}{\lambda}, \qquad \hat{\mathsf{s}}(z^\star_+) \geq \frac{1}{\lambda} + \frac{1}{2}C_2\varepsilon\frac{1}{\lambda} > \frac{1}{\lambda}.$$

In particular, the output $\hat{\lambda}$ of Algorithm 2, being the solution $\hat{\mathsf{s}}(-\hat{\lambda}) = 1/\lambda$, satisfies $z^\star_- \leq -\hat{\lambda} \leq z^\star_+$, so $|\hat{\lambda} - (-z^\star)| \leq \varepsilon$. Finally, by Eq. (42),

$$z^\star = \Psi(1/\lambda) + O(\frac{\lambda}{\sqrt{m}}) = -\tilde{\lambda} + O(\frac{\lambda}{\sqrt{m}}) = -\tilde{\lambda} + O(\lambda\varepsilon)$$

whenever $m \gtrsim 1/\varepsilon^2$ large enough. Since $\tilde{\lambda} \geq \lambda(1 - (1.5)^{-1}) = \frac{1}{3}\lambda = \Omega(\lambda)$, this implies

$$\frac{|\hat{\lambda} - \tilde{\lambda}|}{\tilde{\lambda}} = O(\varepsilon).$$

Thus, Theorem 3.5 is proved.

$\qquad\square$

In the sequel, the following bound on $\mathbb{E}|\hat{\lambda} - \tilde{\lambda}|$ will be useful.

**Lemma D.3.** *Suppose that $m \geq 1.5\mathsf{d}_H(\lambda)$ and $m \geq M$ for large enough $M$. For any $p > 0$,*

$$\mathbb{E}|\hat{\lambda} - \tilde{\lambda}|^p \leq C_p\lambda^p m^{-p/2}. \tag{44}$$

*Proof.* Let $t_0 > 0$ be a small enough constant, such that $[z^\star - \lambda t_0, z^\star + \lambda t_0] \subseteq I^\star$. By the above, $\Pr(|\hat{\lambda} - \tilde{\lambda}| \geq t_0\lambda) = \Pr(-\hat{\lambda} \notin [z^\star - \lambda t_0, z^\star + \lambda t_0]) \leq Ce^{-cm}$. Set $z^\star_\pm(t) = z^\star \pm \lambda t$, so that by the above, $\Pr(\hat{\mathsf{s}}_m(z^\star_+(t)) < 1/\lambda), \Pr(\hat{\mathsf{s}}_m(z^\star_-(t)) > 1/\lambda) \leq Ce^{-cmt^2}$ for all $t \leq t_0$. We have

$\Pr(|\hat{\lambda} - \tilde{\lambda}| \geq t\lambda) = \Pr(-\hat{\lambda} \notin [z^\star_-(t), z^\star_+(t)]) = \Pr(\{\hat{\mathsf{s}}_m(z^\star_+(t)) < 1/\lambda\} \cup \{\hat{\mathsf{s}}_m(z^\star_-(t)) > 1/\lambda\}) \leq 2Ce^{-cmt^2}$. Thus,

$\mathbb{E}|\hat{\lambda} - \tilde{\lambda}|^p = \mathbb{E}[|\hat{\lambda} - \tilde{\lambda}|^p \mathbb{1}_{|\hat{\lambda}-\tilde{\lambda}|\leq\lambda t_0}] + \mathbb{E}[|\hat{\lambda} - \tilde{\lambda}|^p \mathbb{1}_{|\hat{\lambda}-\tilde{\lambda}|>\lambda t_0}] \leq \mathbb{E}[|\hat{\lambda} - \tilde{\lambda}|^p \mathbb{1}_{|\hat{\lambda}-\tilde{\lambda}|\leq\lambda t_0}] + C\lambda^p e^{-cm}$,

and

$\mathbb{E}[|\hat{\lambda} - \tilde{\lambda}|^p \mathbb{1}_{|\hat{\lambda}-\tilde{\lambda}|\leq\lambda t_0}] = \int_0^{t_0} \lambda^p p t^{p-1} \Pr(|\hat{\lambda} - \tilde{\lambda}| \geq \lambda t)dt \lesssim p\lambda^p \int_0^{t_0} t^{p-1}e^{-cmt^2}dt \leq C_p\lambda^p m^{-p/2}$.

And so, $\mathbb{E}|\hat{\lambda} - \tilde{\lambda}|^p \lesssim \lambda\frac{1}{\sqrt{m}} + \lambda e^{-cm} = O(\lambda\frac{1}{\sqrt{m}})$. $\qquad\square$

# E    PROOF OF THEOREM B.1

By definition, if $\tilde{H} \preceq H$ (in PSD order) then $S\tilde{H}S^\top \preceq SHS^\top$, in particular $\lambda_{\min}(SHS^\top) \geq \lambda_{\min}(S\tilde{H}S^\top)$. Choose $\tilde{H} = H(\mu H + I)^{-1}$, noting that $\|\tilde{H}\| \leq 1/\mu$ can be bounded irrespective of $\|H\|$.

We now lower bound $\lambda_{\min}(S\tilde{H}S)$ using a standard net argument, cf. (Vershynin, 2018). For any $\varepsilon > 0$, let $\mathcal{N}_\varepsilon$ be an $\varepsilon$-net of $\mathcal{S}^{m-1}$ of minimum size. One can show that $|\mathcal{N}_\varepsilon| \leq \left(1 + \frac{2}{\varepsilon}\right)^m$. We have

$$\lambda_{\min}(S\tilde{H}S^\top) = \min_{\|x\|=1} x^\top S\tilde{H}S^\top x \geq \min_{x \in \mathcal{N}_\varepsilon} x^\top S\tilde{H}S^\top x - 2\|S\tilde{H}S^\top\|\varepsilon. \tag{45}$$

Note that for fixed $x \in \mathcal{S}^{m-1}$, $y = S^\top x$ has independent, mean zero, sub-Gaussian entries with $\max_{1 \leq i \leq d} \|y_i\|_{\psi_2} = O(1)$ and variance $1/m$.

We first give a high-probability upper bound on $\|S\tilde{H}S^\top\|$. Let $\mathcal{N}_{1/4}$ a $1/4$-net of minimum size, so that

$$\|S\tilde{H}S^\top\| = \max_{\|x\|=1} x^\top S\tilde{H}Sx \leq \max_{x \in \mathcal{N}_{1/4}} x^\top S\tilde{H}Sx + 2 \cdot \|S\tilde{H}S^\top\| \cdot \frac{1}{4},$$

hence $\|S\tilde{H}S^\top\| \leq 2\max_{x \in \mathcal{N}_{1/4}} x^\top S_m\tilde{H}S_m x$. By the Hanson-Wright inequality, Lemma 95 (see also (Vershynin, 2018, Theorem 6.2.1)),

$$\Pr(\|S\tilde{H}S^\top\| \geq 2m^{-1}\operatorname{tr}(\tilde{H}) + t) \leq |\mathcal{N}_{1/4}| \max_{x \in \mathcal{N}_{1/4}} \Pr\left(x^\top S\tilde{H}Sx \geq m^{-1}\operatorname{tr}(\tilde{H}) + t/2\right)$$

$$\leq 9^m 2\exp\left(-c\min\{\frac{m^2 t^2}{\|\tilde{H}\|_F^2}, \frac{mt}{\|\tilde{H}\|}\}\right).$$

Note that

$$\operatorname{tr}(\tilde{H}) = \frac{1}{\mu}\mathsf{d}_H(1/\mu), \qquad \|\tilde{H}\| \leq 1/\mu, \qquad \|\tilde{H}\|_F \leq \sqrt{\|\tilde{H}\|\operatorname{tr}(\tilde{H})} \leq \frac{1}{\mu}\sqrt{\mathsf{d}_H(1/\mu)}.$$

Set $t = C_1 \max\{\frac{1}{\sqrt{m}}\|\tilde{H}\|_F, \|\tilde{H}\|\}$ large enough. For such choice, w.p. $1 - Ce^{-cm}$,

$$\|S\tilde{H}S\| \leq \frac{2}{m}\operatorname{tr}(\tilde{H}) + C_1\max\{\frac{1}{\sqrt{m}}\|\tilde{H}\|_F, \|\tilde{H}\|\}$$

$$\lesssim \frac{1}{\mu}\frac{\mathsf{d}_H(1/\mu)}{m} + \frac{1}{\mu}\max\{1, \sqrt{\frac{d_{1/\mu}}{m}}\} \tag{46}$$

Next, we give a high-probability lower bound on $\min_{x \in \mathcal{N}_\varepsilon} x^\top S\tilde{H}S^\top x$ in (45). Again by Hanson-Wright,

$$\Pr(\min_{x \in \mathcal{N}_\varepsilon} x^\top S\tilde{H}S^\top x \leq m^{-1}\operatorname{tr}(\tilde{H}) - t) \leq \left(1 + \frac{2}{\varepsilon}\right)^m 2\exp\left(-c\min\{\frac{m^2 t^2}{\|\tilde{H}\|_F^2}, \frac{mt}{\|\tilde{H}\|}\}\right),$$

so that w.p. $1 - Ce^{-cm}$, for small $\varepsilon > 0$,

$$\min_{x \in \mathcal{N}_\varepsilon} x^\top S\tilde{H}S^\top x \geq \frac{1}{\mu}\frac{\mathsf{d}_H(1/\mu)(H)}{m} - C_2\log(1 + 1/\varepsilon)\frac{1}{\mu}\max\{1, \sqrt{\frac{\mathsf{d}_H(1/\mu)(H)}{m}}\} \tag{47}$$

Inserting (46)-(47) into (45) implies that w.p. $1 - Ce^{-cm}$,

$$\lambda_{\min}(SHS^\top) \geq (1 - O(\varepsilon))\frac{1}{\mu}\frac{\mathsf{d}_H(1/\mu)}{m} - O(\varepsilon + \log(1 + 1/\varepsilon))\frac{1}{\mu}\max\{1, \sqrt{\frac{\mathsf{d}_H(1/\mu)(H)}{m}}\}. \tag{48}$$

By choosing $\varepsilon > 0$ a sufficiently small numerical constant, we can deduce the following: there are numerical constants $c, C, c_1, c_2 > 0$ such that if $\mathsf{d}_H(1/\mu)/m > 1/c_1$, then w.p. $1 - Ce^{-cm}$, $\lambda_{\min}(\tilde{H}) \geq c_2\frac{1}{\mu}\frac{\mathsf{d}_H(1/\mu)}{m}$. To recover the claimed result, use $\mu = 1/\eta$.

$\square$

# F   PROOF OF LEMMA B.2

The following argument is standard (cf. (Bai & Silverstein, 2010)) and presented for completeness.

Let $r_1, \ldots, r_m \in \mathbb{R}^d$ be the rows of $S$; that is, $S^\top = [r_1 | \ldots | r_m]$. Let

$$\hat{\Sigma}_m = H^{1/2} S^\top S H^{1/2} = \frac{1}{m} \sum_{i=1}^n H^{1/2} r_i r_i^\top H^{1/2} \,. \tag{49}$$

This is a sample covariance matrix, corresponding to $m$ samples with population covariance $\mathbb{E}[\hat{\Sigma}_m] = H$. Recall that $\hat{\Sigma}_m \in \mathbb{R}^{d \times d}$ and $S_m H S_m^\top \in \mathbb{R}^{m \times m}$ have the same non-zero eigenvalues. Denote

$$P_m(z) = (\hat{\Sigma}_m + zI)^{-1} \,, \tag{50}$$

so that

$$\hat{s}(z) = m^{-1} \operatorname{tr}(P_m(z)) + m^{-1}(d - m)\frac{1}{z} \,. \tag{51}$$

Central to the proof is the following leave-one-out decomposition:

$$P_m(z) = P_m^{-k}(z) - \frac{\frac{1}{m} P_m^{-k}(z) H^{1/2} r_k r_k^\top H^{1/2} P_m^{-k}(z)}{1 + \frac{1}{m} r_k^\top H^{1/2} P_m^{-k}(z) H^{1/2} r_k} \,, \tag{52}$$

$$P_m^{-k}(z) = (\hat{\Sigma}_m - \frac{1}{m} H^{1/2} r_k r_k^\top H^{1/2} + zI)^{-1}, \qquad k = 1, \ldots, m \,. \tag{53}$$

The above can be readily shown by the Sherman-Morrison lemma (Lemma M.1).

We shall now prove Lemma B.2 by a standard martingale concentration argument. Let $\mathcal{F}_k$ be the $\sigma$-algebra generated by $r_1, \ldots, r_k$, $k = 0, 1, \ldots, m$, and $\mathbb{E}_{\leq k}[\cdot] := \mathbb{E}[\cdot | \mathcal{F}_k]$. Decompose into a sum of martingale differences,

$$\hat{s}(z) - \mathbb{E}[\hat{s}(z)] = m^{-1} \sum_{k=1}^m (\mathbb{E}_{\leq k} - \mathbb{E}_{\leq k-1})[\operatorname{tr} P_m(z)]$$

$$= m^{-1} \sum_{k=1}^m (\mathbb{E}_{\leq k} - \mathbb{E}_{\leq k-1})[D_{m,k}(z)] \,, \tag{54}$$

where, using (52),

$$D_{m,k}(z) = \frac{\frac{1}{m} r_k^\top H^{1/2} (P_m^{-k}(z))^2 H^{1/2} r_k}{1 + \frac{1}{m} r_k^\top H^{1/2} P_m^{-k}(z) H^{1/2} r_k} \,. \tag{55}$$

Note that $0 \preceq P_m^{-k}(z) \preceq \frac{1}{-z} I$, hence

$$|D_{m,k}(z)| \leq \frac{1}{-z} \cdot \frac{\frac{1}{m} r_k^\top H^{1/2} (P_m^{-k}(z)) H^{1/2} r_k}{1 + \frac{1}{m} r_k^\top H^{1/2} P_m^{-k}(z) H^{1/2} r_k} \leq \frac{1}{-z} \,.$$

Thus, (54) is a sum of bounded martingale differences $|(\mathbb{E}_{\leq k} - \mathbb{E}_{\leq k-1})[D_{m,k}(z)]| \leq \frac{2}{-z}$. To conclude, use the Azuma-Hoeffding inequality (Lemma M.4).

Remark: The above calculation (regarding the boundedness of $D_{m,k}(z)$) is, essentially, the proof of the well-known low-rank perturbation bound for resolvents, Lemma M.3, which we shall use later.

$\square$

## G  PROOF OF THEOREM B.3

We implement a well-known computation technique in random matrix theory (cf. Bai & Silverstein (2010)), while carefully keeping track of the error terms.

We rely explicitly on properties of the Gaussian distribution. Denote the eigendecomposition $H = \sum_{\ell=1}^{d} \tau_\ell v_\ell v_\ell^\top$ and

$$H_{\backslash\ell} = H - \tau_\ell v_\ell v_\ell^\top\,, \qquad s_\ell = Sv_\ell\,.$$

for $\ell = 1, \ldots, d$. Note that $s_1, \ldots, s_d \sim \mathcal{N}(0, m^{-1}I_m)$ are independent (since $v_1, \ldots, v_d$ are orthogonal).

Recall Eq. (51), so that

$$\hat{\mathsf{s}}(z) = -\frac{1}{m}(d-m)\frac{1}{z} + \frac{1}{m}\operatorname{tr}(H^{1/2}S^\top SH^{1/2} - zI)^{-1}$$

$$= -\frac{1}{m}(d-m)\frac{1}{z} + \frac{1}{m}\sum_{\ell=1}^{d} v_\ell^\top(H^{1/2}S^\top SH^{1/2} - zI)^{-1}v_\ell\,. \tag{56}$$

We now compute an expression for $v_\ell^\top(H^{1/2}S_m^\top S_m H^{1/2} - zI)^{-1}v_\ell$, the $\ell$-th diagonal element of the resolvent $(H^{1/2}S_m^\top S_m H^{1/2} - zI)^{-1}$ written in the population covariance eigenbasis $V = [v_1|\ldots|v_d]^\top$.

Upon a coordinate permutation (where $\ell$ becomes the first), we have

$$V^\top(H^{1/2}S^\top SH^{1/2} - zI)V = \begin{bmatrix} \tau_\ell\|s_\ell\|^2 - z & \sqrt{\tau_\ell}s_\ell SH_{\backslash\ell}^{1/2} \\ \sqrt{\tau_\ell}H_{\backslash\ell}^{1/2}S^\top s_\ell & H_{\backslash\ell}^{1/2}S^\top SH_{\backslash\ell}^{1/2} - zI \end{bmatrix}\,.$$

By the block matrix inverse formula (Lemma M.2),

$$v_\ell^\top(H^{1/2}S^\top SH^{1/2} - zI)^{-1}v_\ell = (V^\top(H^{1/2}S^\top SH^{1/2} - zI)V)_{\ell,\ell}^{-1}$$

$$= \left(\tau_\ell\|s_\ell\|^2 - z - \tau_\ell s_\ell SH_{\backslash\ell}^{1/2}(H_{\backslash\ell}^{1/2}S^\top SH_{\backslash\ell}^{1/2} - zI)^{-1}H_{\backslash\ell}^{1/2}S^\top s_\ell\right)^{-1}$$

$$\overset{(\star)}{=} \left(\tau_\ell\|s_\ell\|^2 - z - \tau_\ell s_\ell(SH_{\backslash\ell}S^\top - zI)^{-1}SH_{\backslash\ell}S^\top s_\ell\right)^{-1}$$

$$= -\frac{1}{z}\left(1 + \tau_\ell s_\ell(SH_{\backslash\ell}S^\top - zI)^{-1}s_\ell\right)^{-1}\,, \tag{57}$$

where in $(\star)$ we used the identity $X^\top f(XX^\top)X = f(X^\top X)X^\top X$ which holds for any matrix $X$ and analytic function $f(\cdot)$.

We now wish to estimate the expectation of (57). Denote

$$D_\ell = s_\ell(SH_{\backslash\ell}S^\top - zI)^{-1}s_\ell - \mathbb{E}\hat{\mathsf{s}}(z)$$

$$= D_{\ell,1} + D_{\ell,2} + D_{\ell,3}$$

where

$$D_{\ell,1} = s_\ell(SH_{\backslash\ell}S^\top - zI)^{-1}s_\ell - \frac{1}{m}\operatorname{tr}(SH_{\backslash\ell}S^\top - zI)^{-1}\,,$$

$$D_{\ell,2} = \frac{1}{m}\operatorname{tr}(SH_{\backslash\ell}S^\top - zI)^{-1} - \frac{1}{m}\operatorname{tr}(SHS^\top - zI)^{-1}\,,$$

$$D_{\ell,3} = \frac{1}{m}\operatorname{tr}(SHS^\top - zI)^{-1} - \frac{1}{m}\mathbb{E}\operatorname{tr}(SHS^\top - zI)^{-1}\,.$$

We have

$$v_\ell^\top(H^{1/2}S^\top SH^{1/2} - zI)^{-1}v_\ell = -\frac{1}{z}\left(1 + \tau_\ell\mathbb{E}\hat{\mathsf{s}}(z)\right)^{-1} + \frac{1}{z}\frac{\tau_\ell D_\ell}{\left(1 + \tau_\ell s_\ell(SH_{\backslash\ell}S^\top - zI)^{-1}s_\ell\right)(1 + \tau_\ell\mathbb{E}\mathsf{s}(z))}\,,$$

so that using (56),

$$
\begin{aligned}
\mathbb{E}\mathsf{s}(z) &= \frac{1}{m}(d-m)\frac{1}{z} + \frac{1}{m}\sum_{\ell=1}^{d}\mathbb{E}v_\ell^\top (H^{1/2}S^\top SH^{1/2} - zI)^{-1}v_\ell \\
&= \frac{1}{m}(d-m)\frac{1}{z} - \frac{1}{m}\sum_{\ell=1}^{d}\frac{1}{z}\left(1 + \tau_\ell\mathbb{E}\hat{\mathsf{s}}(z)\right)^{-1} + e_m(z) \\
&= -\frac{1}{z} + \frac{1}{z}\frac{1}{m}\operatorname{tr}\left[I - (I + \mathbb{E}\hat{\mathsf{s}}(z)H)^{-1}\right] + e_m(z) \\
&= -\frac{1}{z}\left(1 - \frac{1}{m}\mathsf{d}_H(1/\mathbb{E}\hat{\mathsf{s}}(z))\right) + e_m(z),
\end{aligned}
\tag{58}
$$

where

$$
e_m(z) = \frac{1}{zm}\sum_{\ell=1}^{m}\mathbb{E}\left[\frac{\tau_\ell D_\ell}{(1 + \tau_\ell s_\ell(SH_{\backslash\ell}S^\top - zI)^{-1}s_\ell)(1 + \tau_\ell\mathbb{E}\mathsf{s}(z))}\right].
$$

We may now bound

$$
\begin{aligned}
|e_m(z)| &\leq \frac{1}{m|z|}\sum_{\ell=1}^{d}\frac{\tau_\ell}{1 + \tau_\ell\mathbb{E}\hat{\mathsf{s}}(z)}\max_{1\leq\ell\leq d}\mathbb{E}|D_\ell| \\
&= \frac{1}{|z|\mathbb{E}\hat{\mathsf{s}}(z)}\frac{1}{m}\mathsf{d}_H(1/\mathbb{E}\hat{\mathsf{s}}(z))\max_{1\leq\ell\leq d}\mathbb{E}|D_\ell|.
\end{aligned}
\tag{59}
$$

We now decompose $\mathbb{E}|D_\ell| \leq \mathbb{E}|D_{\ell,1}| + \mathbb{E}|D_{\ell,2}| + \mathbb{E}|D_{\ell,3}|$. By Lemma M.8 (Item 1),

$$
\mathbb{E}|D_{\ell,1}| \leq \sqrt{\mathbb{E}|D_{1,\ell}|^2} \lesssim \frac{1}{m}\mathbb{E}\|(S_m H_{\backslash\ell}S_m - zI)^{-1}\|_F \leq \frac{1}{|z|\sqrt{m}}.
$$

By the low rank resolvent perturbation lemma, Lemma M.3, almost surely

$$
|D_{\ell,2}| \lesssim \frac{1}{|z|m}.
$$

Finally, by Lemma B.2,

$$
\mathbb{E}|D_{\ell,3}| = \mathbb{E}|\hat{\mathsf{s}}(z) - \mathbb{E}\hat{\mathsf{s}}_m(z)| \lesssim \frac{1}{|z|\sqrt{m}}.
$$

Thus,

$$
|e_m(z)| = O\left(\frac{1}{|z|\mathbb{E}\hat{\mathsf{s}}(z)}\frac{1}{m}\mathsf{d}_H(1/\mathbb{E}\hat{\mathsf{s}}(z))\frac{1}{|z|\sqrt{m}}\right),
\tag{60}
$$

and so the proof is concluded.

$\square$

## H   PROOF OF THEOREM 3.6

Let $\hat{\lambda}$ the output of Algorithm 2, and denote

$$\mathcal{E} = (H + \lambda I)^{1/2}\hat{W}(H + \lambda I)^{1/2} - I, \qquad \hat{W} := S^\top(SHS^\top + \hat{\lambda}I)^{-1}S. \tag{61}$$

We have $\bar{\mathcal{E}} = \frac{1}{q}\sum_{\ell=1}^q \mathcal{E}^{(\ell)}$, where $\mathcal{E}^{(1)}, \dots, \mathcal{E}^{(q)} \overset{\text{i.i.d.}}{\sim} \mathcal{E}$.

The proof proceeds in two parts. First we show that the expectation $\mathbb{E}[\mathcal{E}]$ is small; then, we show that $\bar{\mathcal{E}}$ concentrates around $\mathbb{E}[\bar{\mathcal{E}}] = \mathbb{E}[\mathcal{E}]$ using matrix concentration inequalities.

The following lemma, proven in Section I, bounds $\mathbb{E}[\mathcal{E}]$.

**Lemma H.1.** *Suppose that $S$ has i.i.d. Gaussian entries. Assume that $m \geq 1.5\mathsf{d}_H(\lambda)$. We have*

$$\|\mathbb{E}[\mathcal{E}]\| = O(\frac{1}{\sqrt{m}}) \,.$$

To establish concentration, we use the matrix Bernstein inequality (Tropp et al., 2015, Theorem 6.6.1). We cite it here.

**Lemma H.2.** *Let $X_1, \dots, X_n \in \mathbb{R}^{d\times d}$ be independent Hermitian random matrices. Assume that $\mathbb{E}[X_n] = 0$, $\|X_n\| \leq L$, and denote*

$$\nu^2 = \left\|\sum_{n=1}^N \mathbb{E}[X_n^2]\right\| \,. \tag{62}$$

*Then for all $t \geq 0$,*

$$\Pr\left(\left\|\sum_{n=1}^N X_n\right\| \geq t\right) \leq d\exp\left(\frac{-t^2/2}{\nu^2 + Lt/3}\right) \,. \tag{63}$$

We shall apply Lemma H.2 with a truncated version of $\mathcal{E}$, $\tilde{\mathcal{E}}_L = \mathcal{E}\mathbb{1}_{\|\mathcal{E}\| \leq L} - \mathbb{E}[\mathcal{E}\mathbb{1}_{\|\mathcal{E}\| \leq L}]$ for a suitably chosen $L$. To this end, we give the following high-probability bound on $\|\Phi\|$.

**Lemma H.3.** *Suppose that $S$ has i.i.d. Gaussian entries. For $\delta \in (0,1)$ and $q \geq 1$, set*

$$L_{\delta,q} = C\frac{d}{m} + C\frac{\log(q/\delta)}{m} \tag{64}$$

*for large enough $C > 0$. Then,*

1. *W.p. $1 - \delta/q$, $\|\mathcal{E}\| \leq L_{\delta,q}$.*

2. $\mathbb{E}[\|\mathcal{E}\|\mathbb{1}_{\|\mathcal{E}\|\geq L_{\delta,q}}] = O(\frac{1}{m}e^{-d})$.

3. $\mathbb{E}[\|\mathcal{E}\|^2\mathbb{1}_{\|\mathcal{E}\|\geq L_{\delta,q}}] = O(\frac{d}{m^2}e^{-d})$.

We prove Lemma H.3 in Section J.

The last component needed to apply matrix Bernstein is the variance proxy (62). Note that we always have $\nu^2 \leq L^2 N$, so that

$$\Pr\left(\left\|\frac{1}{N}\sum_{n=1}^N X_n\right\| \geq \varepsilon\right) \leq d\exp(-cN\min\left\{\left(\frac{\varepsilon}{L}\right)^2, \frac{\varepsilon}{L}\right\}) \,. \tag{65}$$

However, if the variance $\mathbb{E}[X^2] \ll L^2$, one may obtain substantially sharper bounds for small $\varepsilon \ll L$. This is the case in our setting. The following is proved in Section K.

**Lemma H.4.** *Suppose that $S$ has i.i.d. Gaussian entries. Assume that $m \geq 1.5\mathsf{d}_H(\lambda)$. Then*

$$\mathbb{E}[\mathcal{E}^2] = O(d/m) \,.$$

With the above, we are ready to conclude the proof of Theorem 3.6.

*Proof.* (Of Theorem 3.6.)

Consider the truncated matrices $\tilde{\mathcal{E}}_{L_{\delta,q}}^{(1)}, \ldots, \tilde{\mathcal{E}}_{L_{\delta,q}}^{(q)}$ with $L_{\delta,q}$ as in Lemma H.3. By Item 1 of Lemma H.3 (and union bound over $\ell = 1, \ldots, q$), with probability $1 - \delta$, we have

$$\bar{\mathcal{E}} = \frac{1}{q} \sum_{\ell=1}^{q} \tilde{\mathcal{E}}_{L_{\delta,q}}^{(\ell)} + \mathbb{E}\left[\mathcal{E}\mathbb{1}_{\|\mathcal{E}\| \leq L_{\delta,q}}\right] . \tag{66}$$

Note that

$$\left\|\mathbb{E}\left[\mathcal{E}\mathbb{1}_{\|\mathcal{E}\| \leq L_{\delta,q}}\right]\right\| = \left\|\mathbb{E}[\mathcal{E}] + \mathbb{E}\left[\mathcal{E}\mathbb{1}_{\|\mathcal{E}\| > L_{\delta,q}}\right]\right\| \leq \|\mathbb{E}[\mathcal{E}]\| + \mathbb{E}\left[\|\mathcal{E}\|\mathbb{1}_{\|\mathcal{E}\| > L_{\delta,q}}\right] = O(1/\sqrt{m}), \tag{67}$$

where the first inequality uses Jensen's inequality, and the second inequality follows from Lemma H.3 Item 2 and Lemma H.1. Similarly,

$$\left\|\mathbb{E}\left[\tilde{\mathcal{E}}_{L_{\delta,q}}^2\right]\right\| \lesssim \left\|\mathbb{E}\left[\mathcal{E}^2\right]\right\| + \mathbb{E}\left[\|\mathcal{E}\|^2\mathbb{1}_{\|\mathcal{E}\| > L_{\delta,q}}\right] = O(d/m), \tag{68}$$

where we used Lemma H.4.

Now, by the matrix Bernstein inequality (Lemma H.2), the bound

$$\|\bar{\mathcal{E}}\| \leq \varepsilon + O(1/\sqrt{m}) \tag{69}$$

holds with probability at least

$$1 - d\exp\left(-cq\min\{\frac{\varepsilon^2}{d/m}, \frac{\varepsilon}{L_{p,q}}\}\right) . \tag{70}$$

When $q \leq \exp(O(d))$ and $1/\delta \leq \exp(O(d))$ we have $L_{\delta,q} = O(d/m)$, so for $\varepsilon = O(1)$ the smaller term is $\frac{\varepsilon^2}{d/m}$. And so, $q \gtrsim \frac{d\log d}{m} \frac{\log(1/\delta)}{\varepsilon^2}$ ensures that the probability above is $\geq 1 - \delta$.

$\square$

# I    PROOF OF LEMMA H.1

We use the machinery from Section G. As before, we denote the eigendecomposition of $H$,

$$H = V \operatorname{diag}(\tau_1, \ldots, \tau_d)V^\top, \qquad V = [v_1|\cdots|v_d]$$

and

$$H_{\setminus\ell} = H - \tau_\ell v_\ell v_\ell^\top, \qquad s_\ell = Sv_\ell, \qquad s_1, \ldots, s_d \overset{\text{i.i.d.}}{\sim} \mathcal{N}(0, m^{-1}I_m).$$

We shall compute the expectation of $\hat{W} = S^\top(SHS^\top + I)^{-1}S$ in the eigenbasis $V$, that is, $\mathbb{E}[V^\top \hat{W} V]$.

Denote, for brevity,

$$T(\hat{\lambda}) = (SHS^\top + \hat{\lambda}I)^{-1}, \qquad T_{\setminus\ell}(\hat{\lambda}) = (SH_{\setminus\ell}S^\top + \hat{\lambda}I)^{-1},$$

so that by the Sherman-Morrison formula (Lemma M.1), writing $SHS^\top = SH_{\setminus\ell}S^\top + \tau_\ell s_\ell s_\ell^\top$,

$$T(\hat{\lambda}) = T_{\setminus\ell}(\hat{\lambda}) - \frac{\tau_\ell T_{\setminus\ell}(\hat{\lambda})s_\ell s_\ell T_{\setminus\ell}(\hat{\lambda})}{1 + \tau_\ell s_\ell T_{\setminus\ell}(\hat{\lambda})s_\ell} . \tag{71}$$

Let $j \neq \ell$. Note that $\hat{\lambda}, T_{\setminus\ell}(\hat{\lambda})$ do not change under a sign flip $s_j \mapsto -s_j$. Since $s_j$ has a symmetric distribution ($s_j \overset{d}{=} -s_j$), we deduce that

$$\mathbb{E}[v_\ell^\top \hat{W} v_j] = \mathbb{E}[s_\ell^\top T(\hat{\lambda})s_j] = 0, \tag{72}$$

that is, $\mathbb{E}[V^\top \hat{W} V]$ is diagonal. Consequently, $\mathbb{E}[V^\top \mathcal{E} V] = \mathbb{E}[V^\top (H + \lambda I)^{1/2}\hat{W}(H + \lambda I)^{1/2}V]$ is diagonal as well (since $V$ is an eigenbasis of $H$).

Let us calculate the diagonal elements. By (71),

$$v_\ell^\top \hat{W} v_\ell = s_\ell^\top T(\hat{\lambda}) s_\ell = \frac{s_\ell^\top T_{\backslash \ell}(\hat{\lambda}) s_\ell}{1 + \tau_\ell s_\ell^\top T_{\backslash \ell}(\hat{\lambda}) s_\ell} = \frac{1}{(s_\ell^\top T_{\backslash \ell}(\hat{\lambda}) s_\ell)^{-1} + \tau_\ell} \,, \tag{73}$$

so

$$
\begin{aligned}
v_\ell^\top \Phi v_\ell &= \frac{\lambda + \tau_\ell}{(s_\ell^\top T_{\backslash \ell}(\hat{\lambda}) s_\ell)^{-1} + \tau_\ell} - 1 \\
&= \frac{\lambda - (s_\ell^\top T_{\backslash \ell}(\hat{\lambda}) s_\ell)^{-1}}{(s_\ell^\top T_{\backslash \ell}(\hat{\lambda}) s_\ell)^{-1} + \tau_\ell} \\
&= O(|\lambda s_\ell^\top T_{\backslash \ell}(\hat{\lambda}) s_\ell - 1|) \,. \tag{74}
\end{aligned}
$$

Let $\tilde{\lambda}$ be given by (9), so that by Theorem 3.5, $\hat{\lambda}$ is close to $\tilde{\lambda}$ with high probability. Let

$$
\begin{aligned}
\Delta_1 &= \lambda s_\ell^\top T_{\backslash \ell}(\hat{\lambda}) s_\ell - \lambda s_\ell^\top T_{\backslash \ell}(\tilde{\lambda}) s_\ell \,, \\
\Delta_2 &= \lambda s_\ell^\top T_{\backslash \ell}(\tilde{\lambda}) s_\ell - \lambda \frac{1}{m} \operatorname{tr} T_{\backslash \ell}(\tilde{\lambda}) \,, \\
\Delta_3 &= \lambda \frac{1}{m} \operatorname{tr} T_{\backslash \ell}(\tilde{\lambda}) - \lambda \frac{1}{m} \operatorname{tr} T(\tilde{\lambda}) \,, \\
\Delta_4 &= \lambda \frac{1}{m} \operatorname{tr} T(\tilde{\lambda}) - \lambda \frac{1}{m} \operatorname{tr} T(\hat{\lambda}) = \lambda \frac{1}{m} \operatorname{tr} T(\tilde{\lambda}) - 1 \,,
\end{aligned}
$$

so that

$$\lambda s_\ell^\top T_{\backslash \ell}(\hat{\lambda}) s_\ell - 1 = \Delta_1 + \Delta_2 + \Delta_3 + \Delta_4 \,.$$

We have

$$\mathbb{E}|\Delta_1| = \mathbb{E}[\lambda |s_\ell^\top T_{\backslash \ell}(\hat{\lambda}) T_{\backslash \ell}(\tilde{\lambda}) s_\ell| |\hat{\lambda} - \tilde{\lambda}|] = \mathbb{E}\left[\frac{\lambda}{\hat{\lambda}\tilde{\lambda}} \|s_\ell\|^2 |\hat{\lambda} - \tilde{\lambda}|\right] \lesssim \frac{1}{\lambda} (\mathbb{E}\|s_\ell\|^4\|)^{1/2} (\mathbb{E}|\hat{\lambda} - \tilde{\lambda}|^2)^{1/2} \lesssim \frac{1}{\sqrt{m}} \,,$$

$$\mathbb{E}|\Delta_3| = \lambda \frac{1}{m} \mathbb{E}[\operatorname{tr}(T_{\backslash \ell}(\hat{\lambda}) T_{\backslash \ell}(\tilde{\lambda})) |\hat{\lambda} - \tilde{\lambda}|] \leq \mathbb{E}\left[\frac{\lambda}{\hat{\lambda}\tilde{\lambda}} |\hat{\lambda} - \tilde{\lambda}|\right] \lesssim \frac{1}{\sqrt{m}} \,,$$

$$\mathbb{E}|\Delta_4| \lesssim \frac{1}{\sqrt{m}}$$

where we used Lemma D.3 and that $\hat{\lambda} = \Omega(\lambda)$ w.p. 1. As for $\Delta_2$, by Lemma M.8 Item 1,

$$\mathbb{E}|\Delta_2| \lesssim \lambda \frac{1}{m} (\mathbb{E}\|T_{\backslash \ell}(\tilde{\lambda})\|_F^2)^{1/2} \lesssim \frac{1}{\sqrt{m}} \,.$$

Thus, we conclude that $\|\mathbb{E}\mathcal{E}\| = \|\operatorname{diag}(\mathbb{E}V^\top \mathcal{E}V)\| = O(\frac{1}{\sqrt{m}})$.

$\square$

## J  PROOF OF LEMMA H.3

Clearly, $\|\mathcal{E}\| \leq 1 + \|(H + \lambda I)^{1/2} \hat{W} (H + \lambda I)^{1/2}\|$. By the inequality $\sqrt{a + b} \leq \sqrt{a} + \sqrt{b}$ (for $a, b \geq 0$), we have $\|(H + \lambda I)^{1/2} (H^{1/2} + \lambda^{1/2} I)^{-1}\| \leq 1$, so

$$\|\Phi\| \leq \|(H^{1/2} + \lambda^{1/2}) \hat{W} (H^{1/2} + \lambda^{1/2})\| \leq 2\|H^{1/2} \hat{W} H^{1/2}\| + 2\lambda \|\hat{W}\| \,,$$

where the second inequality follows from the fact that $\hat{W}$ is PSD, and therefore for any $u, v$, $(u + v)\hat{W}(u + v) = \|u + v\|_{\hat{W}}^2 \leq 2\|u\|_{\hat{W}}^2 + 2\|v\|_{\hat{W}}^2$ where $\|u\|_{\hat{W}}^2 := u^\top \hat{W} u$ is a norm. We have

$$
\begin{aligned}
H^{1/2} \hat{W} H^{1/2} = H^{1/2} S^\top (SHS^\top + \hat{\lambda} I)^{-1} SH^{1/2} &= (H^{1/2} S^\top SH^{1/2} + \hat{\lambda} I)^{-1} H^{1/2} S^\top SH^{1/2} \\
&= I - \hat{\lambda} (H^{1/2} S^\top SH^{1/2} + \hat{\lambda})^{-1} \,,
\end{aligned}
$$

so that $\|H^{1/2}\hat{W}H^{1/2}\| \leq 2$. Moreover, $\|\hat{W}\| = \|S^\top(SHS^\top + \hat{\lambda}I)^{-1}S\| \leq \frac{1}{\hat{\lambda}}\|S^\top S\|$. Recalling that $\hat{\lambda} \geq 5\lambda/12$ (by construction), $\lambda\|\hat{W}\| \leq \frac{12}{5}\|S^\top S\|$. Combining all the above, $\|\mathcal{E}\| \leq 5 + \frac{24}{5}\|S^\top S\|$.

By (Vershynin, 2018, Theorem 4.6.1), for every $t \geq 0$, w.p. $1 - 2e^{-t}$,

$$\|S^\top S\| \leq 1 + C_1 \max\{\mu, \mu^2\} \quad \text{where} \quad \mu = \sqrt{\frac{d}{m}} + \sqrt{\frac{t}{m}}.$$

for $C_1$ large enough. Since $m \leq d$, note that $\max\{\mu, \mu^2\} = \mu^2 \leq 2\frac{d}{m} + 2\frac{t}{m}$. Item 1 follows by setting $t \sim \log(q/\delta)$.

As for Item 2,

$$
\begin{aligned}
\mathbb{E}[\|S^\top S\|\mathbb{1}_{\|S^\top S\|>1+4C_1\frac{d}{m}}] &= \int_{1+4C_1\frac{d}{m}}^\infty \Pr(\|S^\top S\| \geq s)ds \\
&= 2C_1\frac{1}{m}\int_d^\infty \Pr(\|S^\top S\| \geq 1 + 2C_1\frac{d}{m} + 2C_1\frac{t}{m})dt \\
&\lesssim \frac{1}{m}\int_d^\infty e^{-t}dt = \frac{1}{m}e^{-d}.
\end{aligned}
$$

Deducing Item 2 of the lemma is now easy from the above estimate.

Finally, for Item 3,

$$
\begin{aligned}
\mathbb{E}[\|S^\top S\|^2\mathbb{1}_{\|S^\top S\|>1+4C_1\frac{d}{m}}] &= \int_{1+4C_1\frac{d}{m}}^\infty 2s\Pr(\|S^\top S\|^2 \geq s^2)ds \\
&= 2C_1\frac{1}{m}\int_d^\infty \left(1 + 2C_1\frac{d}{m} + 2C_1\frac{t}{m}\right)\Pr(\|S^\top S\| \geq 1 + 2C_1\frac{d}{m} + 2C_1\frac{t}{m})dt \\
&\lesssim \frac{1}{m}\int_d^\infty \left(1 + 2C_1\frac{d}{m} + 2C_1\frac{t}{m}\right)e^{-t}dt = O(\frac{d}{m^2}e^{-d}).
\end{aligned}
$$

$\square$

## K  PROOF OF LEMMA H.4

We continue from Section I, expressing $\mathcal{E}$ in the eigenbasis of $H$.

Recall that $\mathcal{E} = (H + \lambda)^{1/2}S^\top T(\hat{\lambda})S(H + \lambda)^{1/2} - I$. Then starting from (71), we have for any $\ell, j = 1, \ldots, d$,

$$v_\ell^\top \mathcal{E}v_j = (\tau_\ell + \lambda)^{1/2}(\tau_j + \lambda)^{1/2}\frac{s_j T_{\backslash \ell}(\hat{\lambda})s_\ell}{1 + \tau_\ell s_\ell T_{\backslash \ell}(\hat{\lambda})s_\ell} - \mathbb{1}_{\ell=j}. \tag{75}$$

Consequently, if $j \neq k$ then note that $\mathbb{E}[v_j^\top \mathcal{E}v_\ell v_k^\top \mathcal{E}v_\ell] = 0$, since $s_j, s_k$ have a symmetric distribution.

Recall, we are interested in bounding $\|\mathbb{E}[\mathcal{E}^2]\| = \|\mathbb{E}[(V^\top \mathcal{E}V)^2]\|$. We have

$$\mathbb{E}[(V^\top \mathcal{E}V)^2_{j,k}] = \sum_{\ell=1}^d \mathbb{E}[(V^\top \mathcal{E}V)_{j,\ell}(V^\top \mathcal{E}V)_{k,\ell}],$$

which is zero when $j \neq k$; that is, $\mathbb{E}[(V^\top \mathcal{E}V)^2]$ is diagonal. Let us compute the diagonal elements, $j = k$. We have

$$\mathbb{E}[(V^\top \mathcal{E}V)^2_{k,k}] = \mathbb{E}[(V^\top \mathcal{E}V)_{k,k}(V^\top \mathcal{E}V)_{k,k}] + \sum_{\ell \neq k} \mathbb{E}[(V^\top \mathcal{E}V)_{k,\ell}(V^\top \mathcal{E}V)_{k,\ell}]. \tag{76}$$

The first term is

$$\mathbb{E}[(V^\top \Phi V)^2_{k,k}] = \mathbb{E}\left[\left(\frac{\lambda - (s_\ell^\top T_{\backslash \ell}(\hat{\lambda})s_\ell)^{-1}}{(s_\ell^\top T_{\backslash \ell}(\hat{\lambda})s_\ell)^{-1} + \tau_\ell}\right)^2\right]$$

$$= O(|\lambda s_\ell^\top T_{\backslash \ell}(\hat{\lambda})s_\ell - 1|^2),\qquad(77)$$

where we used (74). Similar to Section I, one can show this is $O(1/m)$. For the second term, using (75),

$$\sum_{\ell=1,\ell\neq k}^{d}(V^\top \mathcal{E}V)_{k,\ell}(V^\top \mathcal{E}V)_{k,\ell} = \sum_{\ell=1,\ell\neq k}^{d}(\tau_k + \lambda)(\tau_\ell + \lambda)\frac{(s_k T_{\backslash k}(\hat{\lambda})s_\ell)^2}{(1 + \tau_k s_k^\top T_{\backslash k}(\hat{\lambda})s_k)^2}$$

$$= (\tau_k + \lambda)\frac{1}{(1 + \tau_k s_k T_{\backslash k}(\hat{\lambda})s_k)^2}s_k^\top T_{\backslash k}(\hat{\lambda})\left[\sum_{\ell=1,\ell\neq k}^{d}(\tau_\ell + \lambda)s_\ell s_\ell^\top\right]T_{\backslash k}(\hat{\lambda})s_k.\qquad(78)$$

For small $c' > 0$, let $\Omega$ be the event that $s_k^\top T_{\backslash k}(\hat{\lambda})s_k \geq c'/\lambda$; by similar calculations as in Section I, for some $c, C > 0$, $\Pr(\Omega^c) \leq Ce^{-cm}$. Then

$$\sum_{\ell=1,\ell\neq k}^{d}(V^\top \mathcal{E}V)_{k,\ell}(V^\top \mathcal{E}V)_{k,\ell}\mathbb{1}_\Omega \lesssim \frac{\lambda^2}{\tau_k + \lambda}s_k^\top T_{\backslash k}(\hat{\lambda})\left[\sum_{\ell=1,\ell\neq k}^{d}(\tau_\ell + \lambda)s_\ell s_\ell^\top\right]T_{\backslash k}(\hat{\lambda})s_k\mathbb{1}_\Omega$$

$$\leq \frac{\lambda^2}{\tau_k + \lambda}s_k^\top T_{\backslash k}(\hat{\lambda})\left[\sum_{\ell=1,\ell\neq k}^{d}(\tau_\ell + \lambda)s_\ell s_\ell^\top\right]T_{\backslash k}(\hat{\lambda})s_k,$$

so taking the expectation,

$$\mathbb{E}\left[\sum_{\ell=1,\ell\neq k}^{d}(V^\top \mathcal{E}V)_{k,\ell}(V^\top \mathcal{E}V)_{k,\ell}\mathbb{1}_\Omega\right] \lesssim \frac{\lambda^2}{\tau_k + \lambda}\frac{1}{m}\mathbb{E}\,\mathrm{tr}\left(T_{\backslash k}(\hat{\lambda})\left[\sum_{\ell=1,\ell\neq k}^{d}(\tau_\ell + \lambda)s_\ell s_\ell^\top\right]T_{\backslash k}(\hat{\lambda})\right)$$

$$\lesssim \frac{\lambda}{\tau_k + \lambda}\frac{1}{m}\mathbb{E}\,\mathrm{tr}\left(\left[\sum_{\ell=1,\ell\neq k}^{d}(\tau_\ell + \lambda)s_\ell s_\ell^\top\right]T_{\backslash k}(\hat{\lambda})\right)$$

$$\leq \frac{1}{m}\sum_{\ell=1,\ell\neq k}^{d}(\tau_\ell + \lambda)\mathbb{E}s_\ell^\top T_{\backslash k}(\hat{\lambda})s_\ell.$$

Using (71),

$$\mathbb{E}s_\ell^\top T_{\backslash k}(\hat{\lambda})s_\ell = \mathbb{E}\left[\frac{s_\ell^\top T_{\backslash k,\ell}(\hat{\lambda})s_\ell}{1 + \tau_\ell s_\ell^\top T_{\backslash k,\ell}(\hat{\lambda})s_\ell}\right] \leq \frac{\mathbb{E}[s_\ell^\top T_{\backslash k,\ell}(\hat{\lambda})s_\ell]}{1 + \tau_\ell \mathbb{E}[s_\ell^\top T_{\backslash k,\ell}(\hat{\lambda})s_\ell]},\qquad(79)$$

where we used Jensen's inequality with the concave function $x \mapsto \frac{x}{1+\tau_\ell x}$, $x > 0$. By a calculation similar to Section I, $\mathbb{E}[s_\ell^\top T_{\backslash k,\ell}(\hat{\lambda})s_\ell] = \frac{1}{\lambda} + O(\frac{1}{\lambda\sqrt{m}})$, hence

$$\mathbb{E}s_\ell^\top T_{\backslash k}(\hat{\lambda})s_\ell \leq \frac{\mathbb{E}[s_\ell^\top T_{\backslash k,\ell}(\hat{\lambda})s_\ell]}{1 + \tau_\ell \mathbb{E}[s_\ell^\top T_{\backslash k,\ell}(\hat{\lambda})s_\ell]} \lesssim \frac{1}{\tau_\ell + \lambda}.$$

Consequently,

$$\mathbb{E}\left[\sum_{\ell=1,\ell\neq k}^{d}(V^\top \mathcal{E}V)_{k,\ell}(V^\top \mathcal{E}V)_{k,\ell}\mathbb{1}_\Omega\right] \lesssim \frac{1}{m}\sum_{\ell=1,\ell\neq k}^{d}(\tau_\ell + \lambda)\mathbb{E}s_\ell^\top T_{\backslash k}(\hat{\lambda})s_\ell \lesssim \frac{d}{m}.\qquad(80)$$

Finally, towards bounding the expectation under the complement event $\Omega^c$ note we can write

$$\sum_{\ell=1,\ell\neq k}^{d} (V^\top \mathcal{E}V)_{k,\ell}(V^\top \mathcal{E}V)_{k,\ell} = (\tau_k + \lambda)\frac{1}{(1 + \tau_k s_k T_{\backslash k}(\hat{\lambda})s_k)^2} \sum_{\ell=1,\ell\neq k}^{d} (\tau_\ell + \lambda)(s_\ell^\top T_{\backslash k}(\hat{\lambda})s_k)^2$$

$$\leq (\tau_k + \lambda)\frac{s_k^\top T_{\backslash k}(\hat{\lambda})s_k}{(1 + \tau_k s_k T_{\backslash k}(\hat{\lambda})s_k)^2} \sum_{\ell=1,\ell\neq k}^{d} (\tau_\ell + \lambda)s_\ell^\top T_{\backslash k}(\hat{\lambda})s_\ell$$

$$\leq (\tau_k + \lambda)\frac{s_k^\top T_{\backslash k}(\hat{\lambda})s_k}{1 + \tau_k s_k T_{\backslash k}(\hat{\lambda})s_k} \sum_{\ell=1,\ell\neq k}^{d} (\tau_\ell + \lambda)s_\ell^\top T_{\backslash k}(\hat{\lambda})s_\ell \,, \quad (81)$$

where the first inequality, $(s_\ell^\top T_{\backslash k}(\hat{\lambda})s_k)^2 \leq (s_\ell^\top T_{\backslash k}(\hat{\lambda})s_\ell)(s_k^\top T_{\backslash k}(\hat{\lambda})s_k)$, follows by Cauchy-Schwartz. We have

$$(\tau_k + \lambda)\frac{s_k^\top T_{\backslash k}(\hat{\lambda})s_k}{1 + \tau_k s_k T_{\backslash k}(\hat{\lambda})s_k} = \frac{\tau_k + \lambda}{\tau_k + (s_k^\top T_{\backslash k}(\hat{\lambda})s_k)^{-1}}$$

$$= 1 + \frac{\lambda - (s_k^\top T_{\backslash k}(\hat{\lambda})s_k)^{-1}}{\tau_k + (s_k^\top T_{\backslash k}(\hat{\lambda})s_k)^{-1}}$$

$$\leq 1 + |\lambda(s_k^\top T_{\backslash k}(\hat{\lambda})s_k) - 1| \,.$$

Recall $\Omega$, the event that $\lambda s_k^\top T_{\backslash k}(\hat{\lambda})s_k \geq c'$; we now need to bound $\mathbb{E}[\sum_{\ell=1,\ell\neq k}^{d}(V^\top \mathcal{E}V)_{k,\ell}(V^\top \mathcal{E}V)_{k,\ell}\mathbb{1}_{\Omega^c}]$. Under $\Omega^c$, $0 \leq \lambda s_k^\top T_{\backslash k}(\hat{\lambda}) < c'$, and so $(\tau_k + \lambda)\frac{s_k^\top T_{\backslash k}(\hat{\lambda})s_k}{1+\tau_k s_k T_{\backslash k}(\hat{\lambda})s_k}\mathbb{1}_{\Omega^c} = O(1)$. Plugging this into (81),

$$\mathbb{E}[\sum_{\ell=1,\ell\neq k}^{d} (V^\top \mathcal{E}V)_{k,\ell}(V^\top \mathcal{E}V)_{k,\ell}\mathbb{1}_{\Omega^c}] \lesssim \sum_{\ell=1,\ell\neq k}^{d} (\tau_\ell + \lambda)\mathbb{E}[s_\ell^\top T_{\backslash k}(\hat{\lambda})s_\ell\mathbb{1}_{\Omega^c}] \,. \quad (82)$$

Using (79),

$$(\tau_\ell + \lambda)\mathbb{E}[s_\ell^\top T_{\backslash k}(\hat{\lambda})s_\ell\mathbb{1}_{\Omega^c}] = \mathbb{E}\left[\frac{\tau_\ell + \lambda}{\tau_\ell + (s_\ell^\top T_{\backslash k,\ell}(\hat{\lambda})s_\ell)^{-1}}\mathbb{1}_{\Omega^c}\right]$$

$$\leq \mathbb{E}\left[\left(1 + |\lambda(s_\ell^\top T_{\backslash k,\ell}(\hat{\lambda})s_\ell) - 1|\right)\mathbb{1}_{\Omega^c}\right]$$

$$\lesssim \mathbb{E}[(1 + \|s_\ell\|^2)\mathbb{1}_{\Omega^c}] \lesssim \Pr(\Omega^c) + \sqrt{\mathbb{E}\|s_\ell\|^4 \Pr(\Omega^c)} \lesssim e^{-cm} \,,$$

as $\Pr(\mathcal{E}^c) \lesssim e^{-cm}$. Thus, $\mathbb{E}[\sum_{\ell=1,\ell\neq k}^{d}(V^\top \mathcal{E}V)_{k,\ell}(V^\top \mathcal{E}V)_{k,\ell}\mathbb{1}_{\Omega^c}] \leq de^{-cm}$. And so, we finally conclude

$$\mathbb{E}[\sum_{\ell=1,\ell\neq k}^{d} (V^\top \Phi V)_{k,\ell}(V^\top \Phi V)_{k,\ell}] = \mathbb{E}\sum_{\ell=1,\ell\neq k}^{d} (V^\top \mathcal{E}V)_{k,\ell}(V^\top \mathcal{E}V)_{k,\ell}\mathbb{1}_{\Omega} + \mathbb{E}\sum_{\ell=1,\ell\neq k}^{d} (V^\top \mathcal{E}V)_{k,\ell}(V^\top \mathcal{E}V)_{k,\ell}\mathbb{1}_{\Omega^c}$$

$$\lesssim \frac{d}{m} + de^{-cm} = O(d/m) \,. \quad (83)$$

$\square$

## L    PROOF OF THEOREM 4.1

Define the **Newton decrement** at a point $\theta \in \mathbb{R}^d$:

$$\mathsf{N}(\theta) = \left( (\nabla G(\theta))^\top (\nabla^2 G(\theta))^{-1} (\nabla G(\theta)) \right)^{1/2} . \tag{84}$$

Denote the approximate Newton decrement, where $\bar{W}(\theta)$ is an approximation of the true inverse Hessian:

$$\tilde{\mathsf{N}}(\theta) = \left( (\nabla G(\theta))^\top \bar{W}(\theta) (\nabla G(\theta)) \right)^{1/2} . \tag{85}$$

Note that if $\bar{W}$ is $\eta$-accurate in the sense of Definition 2, then

$$\sqrt{1-\eta}\mathsf{N}(\theta) \leq \tilde{\mathsf{N}}(\theta) \leq \sqrt{1+\eta}\mathsf{N}(\theta) . \tag{86}$$

For self-concordant functions, it is known that the Newton decrement yields an upper bound on the suboptimality gap $G(\theta) - G(\theta^\star)$:

**Lemma L.1.** *If $\mathsf{N}(\theta) < \mathsf{N}_0$ for some numerical constant $\mathsf{N}_0 \leq 0.68$, then*

$$G(\theta) - G(\theta^\star) \leq \mathsf{N}^2(\theta) .$$

*Proof.* See (Boyd & Vandenberghe, 2004, Section 9.6.3), specifically Eq. (95). □

Clearly, by (86), the approximate Newton decrement $\tilde{\mathsf{N}}(\theta)$ provides an upper bound on the optimality gap provided that the inverse Hessian is $\eta$-accurate, albeit a looser one.

The key tool in our analysis are the following estimates, due to (Pilanci & Wainwright, 2015).

**Lemma L.2.** *Operate under the conditions of Theorem 4.1. There exist numerical constants $\Lambda, \nu > 0$ with $\Lambda < 1/16$ such that the following holds. Assume that $0 \leq \eta < 1/2$.*

- *If at an iteration $t$, one has $\mathsf{N}(\theta_{t-1}) \geq \Lambda$, then at the next (approximate) Newton step: $G(\theta_t) - G(\theta^\star) \leq -ab\nu$, where $a, b > 0$ are the parameters for backtracking line search (see Algorithm 3).*

- *If $\mathsf{N}(\theta_{t-1}) \leq \Lambda$ then at the next iteration $\mathsf{N}(\theta_t) \leq \mathsf{N}(\theta_{t-1})$.*

*Proof.* See (Pilanci & Wainwright, 2015, Lemma 6.4). □

Note that Lemma L.2 implies that within at most

$$T_0 = (G(\theta_0) - G(\theta^\star))/(ab\nu) \tag{87}$$

iterations, we are guaranteed to reach $\theta_t$ such that $\mathsf{N}(\theta_t) \leq \Lambda$; furthermore, once we have achieved that, $\mathsf{N}(\theta_{t'}) \leq \Lambda$ holds for every subsequent iteration $t' \geq t$.

The following is the main estimate for the remainder of the analysis:

**Lemma L.3.** *Operate under the conditions of Theorem 4.1, and assume that $\eta < 1/2$. Then*

$$\mathsf{N}(\theta_t) \leq \frac{(1+\eta)\mathsf{N}^2(\theta_{t-1}) + \eta\mathsf{N}(\theta_{t-1})}{\left(1 - (1+\eta)\mathsf{N}(\theta_{t-1})\right)^2} . \tag{88}$$

*Proof.* This result is (Pilanci & Wainwright, 2015, Lemma 6.6). □

One can further simplify (88) assuming that $\mathsf{N}(\theta_{t-1}) < 1/16$. Coarsely lower bounding the denominator and upper bounding the numerator:

$$\mathsf{N}(\theta_t) \leq 2\mathsf{N}^2(\theta_{t-1}) + 2\eta\mathsf{N}(\theta_{t-1}) \leq \begin{cases} 4\mathsf{N}^2(\theta_{t-1}) & \text{if} \quad \mathsf{N}(\theta_{t-1}) \geq \eta, \\ 4\eta\mathsf{N}(\theta_{t-1}) & \text{if} \quad \mathsf{N}(\theta_{t-1}) \leq \eta \end{cases} , \tag{89}$$

provided that $\mathsf{N}(\theta_{t-1}) < 1/16$.

With the above estimates in hand, we are ready to prove Theorem 4.1.

*Proof.* (Of Theorem 4.1.) Recall that we assume that $\eta < 1/5$, and in particular $\eta < 1/2$. By Lemma L.2, within $T_0$ iterations, the approximate Newton method reaches $\theta_t$ such that $\mathsf{N}(\theta_{t'}) < 1/16$ for all $t' \geq t$. In particular, also $G(\theta) - G(\theta^\star) \leq (1+\eta)1/16^2 \leq 3/500$. Let $\varepsilon > 0$ be the desired precision; suppose that $\varepsilon < 3/500$. We consider two cases.

First, assume that $\eta \leq \varepsilon < 3/500$. Let $T_1(\varepsilon)$ be the smallest integer $t$ such that the dynamic

$$A_t = 4A_{t-1}^2, \quad A_0 = 1/16 \tag{90}$$

satisfies $A_t \leq \varepsilon$. It is easy to verify that $A_t = 1/4(1/4)^{2^t}$ (e.g. by induction), hence $T_1(\varepsilon) = O(\log\log(1/\varepsilon))$. We conclude that when $\varepsilon \geq \eta$, $\mathsf{T}(\varepsilon) \leq T_0 + T_1(\varepsilon)$.

The second case is when $\varepsilon < \eta$, therefore we are in the regime of (89) where decay is linear rather than quadratic. Let $T_2(\varepsilon)$ be the smallest integer $t$ such that the dynamic

$$B_t = 4\eta B_{t-1}, \quad B_0 = \eta \tag{91}$$

satisfies $B_t \leq \varepsilon$. Clearly $B_t = (4\eta)^t B_0$, hence $T_2(\varepsilon) = O(\frac{\log(\eta/\varepsilon)}{\log(1/\eta)})$ (here we used that $4\eta < 1$, by assumption). We conclude that when $\varepsilon < \eta$, $\mathsf{T}(\varepsilon) \leq T_0 + T_1(\eta) + T_2(\varepsilon)$.

$\square$

# M  AUXILLIARY TECHNICAL RESULTS

**Lemma M.1** (Sherman-Morrison). *Let $A \in \mathbb{R}^{n \times n}$ be invertible, and $u, v \in \mathbb{R}^n$.*
*The matrix $A + uv^\top$ is invertible if and only if $1 + v^\top A^{-1} u \neq 0$. In that case,*

$$(A + uv^\top)^{-1} = A^{-1} - \frac{A^{-1} u v^\top A^{-1}}{1 + v^\top A^{-1} u} . \tag{92}$$

**Lemma M.2** (Block matrix inversion). *Let $A, B, C, D$ be matrices of conforming dimensions. Provided that $A, D$ are both invertible,*

$$\begin{bmatrix} A & B \\ C & D \end{bmatrix}^{-1} = \begin{bmatrix} (A - BD^{-1}C)^{-1} & 0 \\ 0 & (D - CA^{-1}B)^{-1} \end{bmatrix} \begin{bmatrix} I & -BD^{-1} \\ -CA^{-1} & I \end{bmatrix} . \tag{93}$$

**Lemma M.3.** *(Resolvent low-rank perturbation.) Let $A, B, C \succeq 0$ be PSD matrices, and $z > 0$. Then*

$$\left| \operatorname{tr} C(zI + A)^{-1} - \operatorname{tr} C(zI + B)^{-1} \right| \leq \operatorname{rank}(A - B) \frac{\|C\|}{z} .$$

For reference, see for example (Bai & Silverstein, 2010).

Let $\{\emptyset, \Omega\} =: \mathcal{F}_0 \leq \mathcal{F}_1 \leq \mathcal{F}_2 \leq \ldots$ be a filtration of $\sigma$-algebras over some given probability space $\Omega$. Recall that a random process $\{X_n\}_{n=1}^\infty$ is a *martingale difference* (adapted to this filtration) if 1) for every $n$, $X_n$ is $\mathcal{F}_n$-measurable; and 2) $\mathcal{F}_{n-1}$-almost surely, $\mathbb{E}[X_n | \mathcal{F}_{n-1}] = 0$.

**Lemma M.4** (Azuma-Hoeffding). *Suppose that $(X_n, \mathcal{F}_n)_{n=1}^N$ is a martingale difference, and $\{B_n\}_{n=1}^N$ constants. Suppose that for all $1 \leq n \leq N$, almost surely, $|X_n| \leq B_n$. Then for all $t \geq 0$,*

$$\Pr(|\sum_{n=1}^N X_n| \geq t) \leq 2 \exp\left( \frac{-\frac{1}{2} t^2}{\sum_{n=1}^N B_n^2} \right) .$$

The sub-Gaussian and sub-Exponential norms of a (scalar) random variable are, respectively

$$\|X\|_{\psi_2} = \inf\left\{ \sigma > 0 \ : \ \mathbb{E} e^{X^2/\sigma^2} \leq 2 \right\} , \qquad \|X\|_{\psi_1} = \inf\left\{ \sigma > 0 \ : \ \mathbb{E} e^{|X|/\sigma} \leq 2 \right\} . \tag{94}$$

We call $X$ sub-Gaussian, resp. sub-Exponential, if $\|X\|_{\psi_2} < \infty$, resp. $\|X\|_{\psi_1} < \infty$.

For a random vector $r$, we define $\|r\|_{\psi_i} = \sup_{\|v\|=1} \|v^\top r\|_{\psi_i}$. That is, it is the largest $\psi_i$-norm of a one-dimensional projection of $r$.

**Lemma M.5.** *The following holds.*

1. *$\| \cdot \|_{\psi_i}$ is a norm on the subspace of random variables $X$ such that $\|X\|_{\psi_i} < \infty$.*

2. *If $X, Y$ are sub-Gaussian (possible statistically dependent), then $XY$ is sub-Exponential, with $\|XY\|_{\psi_1} \leq \|X\|_{\psi_2} \|Y\|_{\psi_2}$.*

3. *If $|Y| \leq B$ almost surely for $B \geq 0$ constant, $\|XY\|_{\psi_i} \leq B\|X\|_{\psi_i}$.*

4. *(Centralization.) For any $X$, $\|X - \mathbb{E}[X]\|_{\psi_i} \leq \|X\|_{\psi_i}$.*

We refer to the book (Vershynin, 2018) for reference.

**Lemma M.6** (Bernstein). *Suppose that $(X_n, \mathcal{F}_n)_{n=1}^N$ is a martingale difference. Then for all $t \geq 0$,*

$$\Pr(|\sum_{n=1}^N X_n| \geq t) \leq 2 \exp\left( -c \min\{ \frac{t}{\max_{n=1,\ldots,N} \|X_n\|_{\psi_1}}, \frac{t^2}{\sum_{n=1}^N \|X_n\|_{\psi_1}^2} \} \right) .$$

Next, we cite the following version of the Hanson-Wright inequality, see (Vershynin, 2018).

**Lemma M.7** (Hanson-Wright). *Suppose that $r$ has independent sub-Gaussian entries, with $\mathbb{E}[r_i] = 0, \mathbb{E}[r_i^2] = 1, \max_i \|r_i\|_{\psi_2} \leq K$. For any matrix $A$ and $t \geq 0$,*

$$\Pr(|r^\top A r - \operatorname{tr}(A)| \geq t) \leq \exp\left(-c \min\{\frac{t}{K^2 \|A\|}, \frac{t^2}{K^4 \|A\|_F^2}\}\right), \tag{95}$$

*where $c > 0$ is a universal constant.*

Lastly, the following lemma collects, for convenience, some properties satisfied by any isotropic random vector $r$ that satisfies a concentration inequality of the form (95).

**Lemma M.8.** *Suppose $r$ is istropic $\mathbb{E}[r] = 0, \mathbb{E}[rr^\top] = I$, and satisfies (95) for some $K > 0$.*

1. *For any matrix $A$ (independent of $r$), $\mathbb{E}[|r^\top A r - \operatorname{tr}(A)|^2] \lesssim K^4 \|A\|_F^2$.*

2. *For any vector $v$ (independent of $r$), $\|v^\top r\|_{\psi_2} \lesssim K\|v\|$. That is, $r$ is sub-Gaussian, with $\|r\|_{\psi_2} \lesssim K$.*

*Proof.* Start with the first item. Write $\mathbb{E}[|r^\top A r - \operatorname{tr}(A)|^2] = \int_0^\infty \Pr(|r^\top A r - \operatorname{tr}(A)|^2 > s) ds = 2\int_0^\infty t \Pr(|r^\top A r - \operatorname{tr}(A)|^2 > t) dt$. Using (95), we crudely bound

$$\int_0^\infty t \Pr(|r^\top A r - \operatorname{tr}(A)|^2 > t) dt \leq \int_0^\infty t \exp(-c\frac{t^2}{K^4 \|A\|_F^2}) dt + \int_0^\infty t \exp(-c\frac{t}{K^2 \|A\|}) dt \lesssim K^4 \|A\|_F^2.$$

As for the second item, note that $\|vv^T\| = \|vv^T\|_F = \|v\|^2$. Hence,

$$\Pr(|v^\top r| \geq t) = \Pr(r^\top vv^\top r \geq t^2) \lesssim \exp(-\min\{\frac{t^2}{K^2 \|v\|^2}, \frac{t^4}{K^4 \|v\|^4}\}).$$

That is, $\|v^\top r\|$ has a Gaussian tail with variance proxy $K^2 \|v\|^2$. $\qquad\square$

