# OpenReview forum: "Newton Meets Marchenko-Pastur: Massively Parallel Second-Order Optimization with Hessian Sketching and Debiasing"
_ICLR.cc/2025/Conference — ICLR 2025 Poster_

### Official Review · Reviewer_1hmD · 2024-10-28

**Soundness:** 3
**Presentation:** 2
**Contribution:** 3
**Rating:** 6
**Confidence:** 2

**Summary:**

This paper propoed a distributed Newton method, where the server performs the Newton update,  and the workers use Hessian skeching to return a neatly unbiased inverse Hessian to the server. The solution is based on random matrix theory, i.e. the Marchenko-Pastur law.

**Strengths:**

The solution in this paper is novel to me. I think the idea is very interesting.

**Weaknesses:**

See the questions part.

**Questions:**

1. The paper says "The connection to the Marchenko-Pastur law treats in a uniform  framework—and recovers almost immediately—earlier results on Hessian sketch debiasing for distributed Newton methods (Derezinskietal.,2020;Zhang&Pilanci,2023). Can the authors compare the works of  (Derezinskietal.,2020;Zhang&Pilanci,2023) and this work for me to better clarify the contributions of this paper.

2. Why the authors claim their result on non-asymptotic error guarantees in Section 3.1 is dimension-free. I can not uderstand this because Theorem 3.6 requires $q = \Omega (d / (m \epsilon^2)) $. I can understand that if we have enough workers, i.e, $q = \Omega (d / (m \epsilon^2)) $ then we can set $m = O(1/\epsilon^2)$, which is dimension-free. But in applications, I think $q$ is considered to be fixed and then to ensure the error is smaller than $\epsilon$, one should take $m = O( d / (q \epsilon^2 )$ if $q <d$. In this view, the gurantee seems to depend on dimension $d$.

3. (continued my second question) What is the total complexity to minimize a self-concordant function? By Theorem4.1 if we have $\eta < \epsilon$, then the algorithm can recovers the $O(\log \log (1/\epsilon))$ rate of Newton's method. By according to Theorem 3.6, $m$ should be $m = O(  1/ \epsilon^2 \lor d / (q \epsilon^2 ))$, which may be very large. The advantage of Newton's method is that it can have superlinear convergence rate, but as $m = \Omega(1/\epsilon^2)$, the total complexity has polynomial dependency on $1/\epsilon$. Then the total complexity results seems to be sublinear.  If so, what is the advatange of Newton's method compared to gradient methods? In other words, can the authors compare the total complexity of their Newton's method and distributed gradient methods (including the accleration) for me?

---

### Official Review · Reviewer_UJF8 · 2024-11-01

**Soundness:** 3
**Presentation:** 3
**Contribution:** 3
**Rating:** 6
**Confidence:** 3

**Summary:**

This paper proposes a novel sketching Newton method. This paper also provides the convergence analysis of the algorithm and gives the non asymptotic guarantees.

**Strengths:**

This paper proposes a novel sketching Newton method. This paper also provides the convergence analysis of the algorithm and gives the non asymptotic guarantees. The massively parallel fashion is consider in their method.
The theory in this paper is good.

**Weaknesses:**

This paper claims that the proposed method is motived by serverless cloud computing, in particular the ``function as a service'' (FaaS) model. However, Eq. (7) is an biased estimation of $W_t$. Thus,  even $m$ machines run in parallel. This will not help to obtaim an $m$-times speed-up.
The convergence rate of the proposed algorithm with $m$ machines running in parallel is the same to one only runing in  a single machine.

The experiments prove the biases of the proposed algorithm is lower than other agorithm and the algorithm can achieve lower iteration complexity.
However, the coresponding computation cost is not mentioned. Thus, the comparisons in this paper seem not enough to prove the advantages of the proposed algorithm.

**Questions:**

No

---

### Official Review · Reviewer_wj6U · 2024-11-02

**Soundness:** 3
**Presentation:** 2
**Contribution:** 2
**Rating:** 6
**Confidence:** 4

**Summary:**

The key idea of the paper is to relate random matrices and one of the key optimization methods, Newton's method. Convergence for strongly convex self condordant functions is proved.

**Strengths:**

1) A new (best of my knowledge) matrix sketching scheme is presented.
2) The proofs look correct.

**Weaknesses:**

1) Convergence only for self condordant functions. The result would be more solid if the authors consider general strongly convex functions.

2) There is no comparison with other methods in the experiments. There is a huge family of Newton-like methods where the Hessian is approximated. Starting from quasi-Newton technique to newer Hutchinson [1] or Fisher approximations [2]. The same Adam can be considered as a method with a diagonal approximation of the Hessian. I could also mention Shampoo or Soap or [5]

[1] Jahani, M., Rusakov, S., Shi, Z., Richt´arik, P., Mahoney, M. W., and Tak´aˇc, M. (2021). Doubly adaptive scaled algorithm for machine learning using second-order information.

[2] Amari, S. (2021). Information geometry.

[3] Gupta, V., Koren, T., and Singer, Y. (2018). Shampoo: Preconditioned stochastic tensor optimization.

[4] Vyas, N., Morwani, D., Zhao, R., Shapira, I., Brandfonbrener, D., Janson, L., and Kakade, S. (2024). Soap: Improving and stabilizing shampoo using adam

[5] Doikov, N., Jaggi, M., et al. (2023). Second-order optimization with lazy hessians.

3) There's an important point in the proof of Theorem 3.6: $\delta$ is small probability, what do we need to do if $d \cdot \log\delta^{-1} \cdot e^{-2}$ is greater than $\delta \cdot e^d$? Maybe I misunderstood something.

**Questions:**

See 3 in Weaknesses

---

### Official Review · Reviewer_K6ze · 2024-11-04

**Soundness:** 3
**Presentation:** 3
**Contribution:** 3
**Rating:** 8
**Confidence:** 3

**Summary:**

The paper presents a new optimization method for distributed centralized optimization, where all workers have access to the full data. The method is based on damped Newton method and inexact approximation of inverse Hessian with sketching by Marchenko-Pastur scheme. The authors present extensive theoretical support of their method with a focus on debiasing properties of the proposed approximation.

**Strengths:**

I think that the results are new. I appreciate the proposed approximation and guarantees by itself without the need of a distributed setting. This approach is fresh for the optimization community. The paper is complicated but well-written.

**Weaknesses:**

1. One of the main weaknesses from my point of view is the experiments section.
a) Firstly, there is no comparison with other methods. So, it is unclear how fast is the proposed method. For example, it can be interesting how far is it from the exact Newton method.
b) Secondly, the Ridge Regression seems to be a very simple task for Newton-based methods, as it is a quadratic function with a relatively small noise($\sigma = 0.01$) and big regulariser $\lambda = 0.001$.  So, it might be much more useful to show how good is the inverse approximation depending on the conditioning of the Hessian and size of $m$, similar to Fig.1.

2. Small remark: I recommend changing the indexing in formulas 2, 3, 4. As $g_t$ is a gradient in point $\theta_{t-1}$ and should be $g_{t-1}$, also Hessian $H_t$ is calculated in $\theta_{t-1}$.

3. Also, the authors may improve the related literature section by some distributed Newton and Quasi-Newton methods. Some possible references:

 [1] Safaryan, M. et.al. "FedNL: Making Newton-type methods applicable to federated learning." (2021) arXiv preprint arXiv:2106.02969.

[2] Agafonov, A. et.al. "Flecs: A federated learning second-order framework via compression and sketching". (2022) arXiv preprint arXiv:2206.02009.

[3] Qian, X. et.al  "Basis matters: better communication-efficient second order methods for federated learning." (2021) arXiv preprint arXiv:2111.01847.

[4] Elgabli, A. et.al. "FedNew: A communication-efficient and privacy-preserving Newton-type method for federated learning". (2022) In International conference on machine learning (pp. 5861-5877). PMLR.

[5] Bischoff, S. et.al. "On second-order optimization methods for federated learning." (2021) arXiv preprint arXiv:2109.02388.

**Questions:**

1) Could you clarify the distributed setting paradigm? As I understand, all workers have access to the whole function and they have the same gradient. As this setting is quite unusual to me, especially from Federated Learning and privacy perspective.

2) Does this method and approximation work for pure convex functions without regularisation? Also, may they work for non-convex functions in practice?

3) How good is the proposed approximation compared to Krylov and Nystorm-based approximations?

4) It might be interesting to apply this approximation to the new branch of methods based on cubic regularisation:

 [6] Scieur, Damien. "Adaptive Quasi-Newton and Anderson acceleration framework with explicit global (accelerated) convergence rates." International Conference on Artificial Intelligence and Statistics. PMLR, 2024

[7] Kamzolov, Dmitry, et al. "Cubic Regularization is the Key! The First Accelerated Quasi-Newton Method with a Global Convergence Rate of $ O (k^{-2}) $ for Convex Functions." arXiv preprint arXiv:2302.04987 (2023)

[8] Jiang, Ruichen, et al. "Krylov Cubic Regularized Newton: A Subspace Second-Order Method with Dimension-Free Convergence Rate." International Conference on Artificial Intelligence and Statistics. PMLR, 2024.

---

### Meta-Review · Area_Chair_vD8A · 2024-12-22

**Metareview:**

The paper presents a new optimization method for distributed centralized optimization, where all workers have access to the full data. The method is based on damped Newton method and inexact approximation of inverse Hessian with sketching by Marchenko-Pastur scheme. The authors present extensive theoretical support of their method with a focus on debiasing properties of the proposed approximation.

All 4 reviewers (scores 8, 8, 6, 6) agreed that the paper should be accepted. I have read the reviews, the rebuttals, and the discussion. Moreover, I have had a quick read of the main body of the paper. I agree that the paper develops some very interesting new results, with potential for further extensions.

- The method and the theoretical results are new and interesting.
- The paper is well written.
- Related work is (after rebuttal) in a good shape.

I recommend the paper for acceptance.

**Additional Comments On Reviewer Discussion:**

Several reviewers raised some concerns and asked some questions; for example, concerns w.r.t. non-cited related literature (resolved by the authors updating their paper), question on the possibility of extension to nonconvex regime (addressed satisfactorily in a reply), comparison to Krylov and Nystrom approximations (addressed by a reply), and more.

In summary, the rebuttal was effective, and the answers satisfactory. Moreover, some of the issues raised were addressed by the authors updating their paper, which is now a bit more complete as a result.

---

### Decision · Program_Chairs · 2025-01-22

Accept (Poster)